# SAPIENS2

**Rawal Khirodkar, He Wen, Julieta Martinez, Yuan Dong, Su Zhaoen, Shunsuke Saito**
Meta Reality Labs
https://github.com/facebookresearch/sapiens2

## ABSTRACT

We present Sapiens2, a model family of high-resolution transformers for human-centric vision focused on generalization, versatility, and high-fidelity outputs. Our model sizes range from $0.4$ to $5$ billion parameters, with native 1K resolution and hierarchical variants that support 4K. Sapiens2 substantially improves over its predecessor in both pretraining and post-training. First, to learn features that capture low-level details (for dense prediction) and high-level semantics (for zero-shot or few-label settings), we combine masked image reconstruction with self-distilled contrastive objectives. Our evaluations show that this unified pretraining objective is better suited for a wider range of downstream tasks. Second, along the data axis, we pretrain on a curated dataset of $1$ billion high-quality human images and improve the quality and quantity of task annotations. Third, architecturally, we incorporate advances from frontier models that enable longer training schedules with improved stability. Our 4K models adopt windowed attention to reason over longer spatial context and are pretrained with 2K output resolution. Sapiens2 sets a new state-of-the-art and improves over the first generation on pose ($+4$ mAP), body-part segmentation ($+24.3$ mIoU), normal estimation ($45.6\%$ lower angular error) and extends to new tasks such as pointmap and albedo estimation.

## 1 INTRODUCTION

Sapiens introduced a foundation model for human-centric vision (Khirodkar et al., 2024). The over-arching goal is to build models that operate across *any* human task and *any* human imagery while maintaining *highest* output fidelity. In this work, we present SAPIENS2, which advances this objective along all three axes—task, image, and fidelity.

*Any human task.* Sapiens primarily relied on MAE (He et al., 2022) pretraining, a form of masked image modeling (MIM) (Hondru et al., 2025). MIM preserves signal and spatial details by optimizing reconstruction and thus primarily learns by compression (Zhang et al., 2022). Unlike language—where tokens are discrete and largely self-semantic and masked modeling has become a default—visual semantics are denser, context-dependent and under-constrained by pixel prediction alone; consequently, MIM features often require moderate-to-high supervision to express semantics reliably. In contrast, contrastive learning (CL) (Chen et al., 2020a) injects semantics by enforcing instance-level invariances using positives and negatives (Chen et al. (2020b), Chen et al. (2021)), yet its global invariance objectives tend to underperform on dense prediction, where fine spatial detail and photometric fidelity matter. This gap has motivated hybrids that combine global CL and MIM - such as iBOT's masked student–teacher matching (Zhou et al., 2021) and successors such as DINOv2 (Oquab et al., 2023) and v-JEPA (Bardes et al., 2024). While these approaches narrow the gap, performance at high resolution remains mixed and can exhibit *representation drift*: aggressive invariances (notably appearance augs.) decouple teacher and student from the true observations, eroding cues—such as color—that are critical for human-centric dense tasks (*e.g.* photorealistic avatar creation). SAPIENS2 addresses these limitations by coupling a reconstruction objective with contrastive objectives, anchoring features in pixel space (Huang et al., 2023) while organizing them semantically. The result is a general-purpose representation that transfers across zero-shot, few-shot (Song et al., 2023), and fully supervised regimes and a broad spectrum of human-centric tasks.

*Any human image.* Generalization scales with data and model capacity. During pretraining, we curate *1B* high-quality human images from a web-scale corpus via multi-stage filtering. The collection spans diverse ages, ethnicities, backgrounds, and real-world conditions, subject to a single constraint: each image contains at least one prominent person. Beyond this human-centric requirement, we use no task labels and inject no human-specific priors during pretraining. For

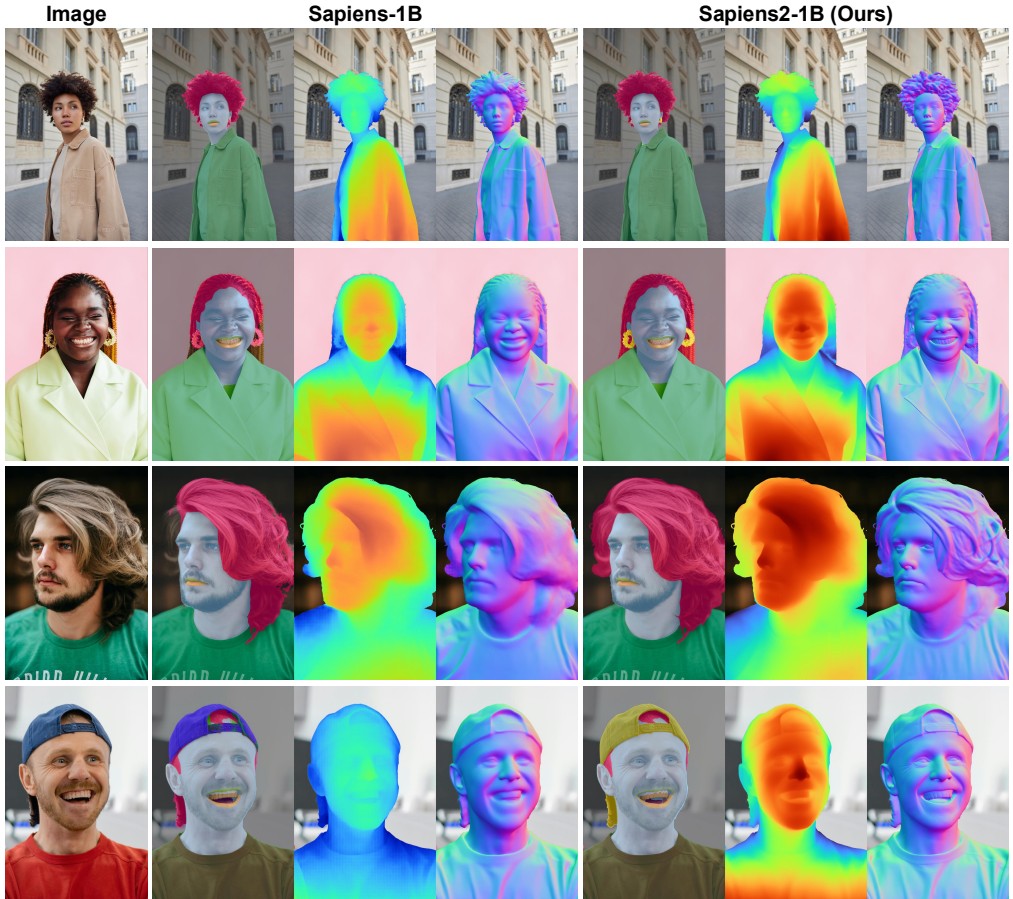

Figure 1: **SAPIENS2 for dense-prediction tasks.** We compare 1B models from both generations on segmentation, depth, and normals. Sapiens2 improves over Sapiens with stronger generalization and sharper segmentation of rare classes (lips, tongue, earrings), achieving pixel-accurate hair segmentation. On geometric tasks (depth, normals), it captures subtler facial, clothing, and hair details—all without task-specific architectures.

post-training, we target fundamental human tasks—pose estimation (Zheng et al., 2023), body-part segmentation Thisanke et al. (2023), surface-normal (Bae & Davison, 2024), pointmap (per-pixel XYZ) (Wang et al., 2024) and albedo estimation (Ran et al., 2024). Relative to Khirodkar et al. (2024), we scale task-specific supervision by $10\times$, typically on the order of 1M labels per task, and improve synthetic assets with more detailed geometry and photorealism. On the model axis, our largest variant has 5B parameters, accompanied by 0.4B, 0.8B, and 1B models for different compute settings and broader use. At a native resolution of 1K, our largest model achieves among the highest FLOPs reported for vision transformers. Fig. 1 showcases improvements over Sapiens for segmentation, depth and normals. Our models segment tiny accessories such as chains and earrings, and separate teeth and gums with pixel accuracy. Additionally, the predicted normals better capture facial wrinkles and hair details. Our evaluations show that learning at scale yields strong generalization across unconstrained human images and challenging in-the-wild conditions.

*Highest fidelity.* Prediction fidelity scales with the number of visual tokens a model processes, which in turn grows with input resolution (Zhao et al., 2018). Beyond standard 1K backbones (Khirodkar et al., 2024), we introduce a 4K backbone pretrained and post-trained for dense prediction, with task heads that decode to 2K resolution across tasks. To make 4K tractable, we adopt a hierarchical design (Li et al., 2022): an initial stack of windowed self-attention layers operates locally to capture texture and fine boundaries, from each window we pool a summary token and then apply global self-attention—mirroring our 1K models—to fuse long-range context. This layout is naturally compatible with MAE-style pretraining: after the local stage, masked tokens can be dropped so that information does not flow across masked regions, avoiding the leakage that convolutional backbones typically require masked convolutions to prevent (Gao et al., 2022). We additionally incorporate targeted efficiency and stability upgrades—RMSNorm in place of LayerNorm (Meta, 2025), grouped-query attention for higher throughput (Ainslie et al., 2023), QK-Norm for robust high-resolution training (Henry et al., 2020)—and employ a pixel-shuffle (Shi et al., 2016) decoder for sub-pixel reasoning. Together, these choices fully exploit our high-resolution setting while keeping memory in check.

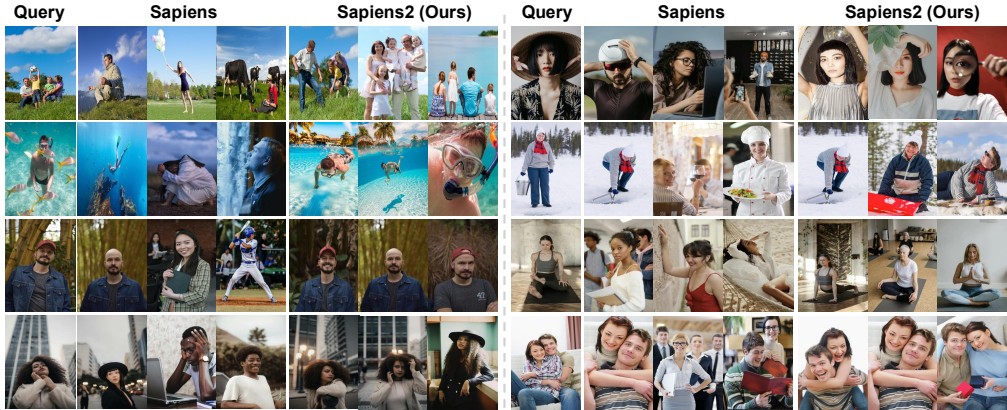

Figure 2: **k-NN comparison using [CLS] token**. SAPIENS2 learns a more discriminative, human-semantic feature space—grouping visually similar concepts and improving retrieval performance at high resolution.

We extensively evaluate SAPIENS2 across various tasks and benchmarks. Figure 2 qualitatively visualizes nearest neighbors retrieved using [CLS] tokens from 1K-resolution Sapiens and SAPIENS2. Our contrastive pretraining yields a feature space that captures human semantics and returns plausible neighbors. Figure 3 further shows that, without any supervision, our model produces human-centric attention maps. Overall, our contributions are summarized as follows.

- SAPIENS2 is a family of transformers (0.4B–5B parameters) pretrained on 1 billion high-quality human images. Our models support 1K native resolution and 4K hierarchical resolution and are designed for high-resolution dense predictions.
- We use masked reconstruction with contrastive objectives to learn features that generalize in zero-shot settings on human tasks while preserving fine details in dense predictions.
- We fine-tune with high-quality annotations for pose, part segmentation, pointmaps, normals, and albedo, achieving state-of-the-art performance across benchmarks.

## 2 RELATED WORK

**Self-Supervised Learning.** Recent breakthroughs in self-supervised learning at scale fall into two families: (1) Masked Image Modeling (MIM) and (2) Contrastive Learning (CL). MIM follows masked language modeling in NLP, but unlike language—where tokens are self-semantic—image patches are context-dependent. Visual representations are thus denser and more ambiguous. MIM objectives are commonly viewed as a form of compression (Zhang et al., 2022) of the input tokens. Among popular approaches, BEiT (Bao et al., 2021) uses a dVAE tokenizer to discretize image patches and trains the model to predict the codebook indices of masked patches, while MAE (He et al., 2022) masks a large fraction of patches (75%) and reconstructs the missing pixels directly. Numerous studies adopt this paradigm for pretraining—e.g., U-MAE, CAE, SiamMAE, MR-MAE, and Sapiens (Khirodkar et al., 2024). Representative methods in CL include BYOL (Grill et al., 2020), SimCLRv2 (Chen et al., 2020b), MoCov3 (Chen et al., 2021), and DINO (Caron et al., 2021). Given their complementarity, combining the objectives is natural; for instance, iBOT (Zhou et al., 2021) combines MIM with CL-style self-distillation, aligning student and teacher features via the masked objective rather than reconstructing pixels or codewords, consistent with JEPA (Assran et al., 2023) and v-JEPA2 (Assran et al., 2025). DINOv2 (Oquab et al., 2023) adopts the iBOT objective as their primary pretraining strategy. DINOv3 (Siméoni et al., 2025) further scales this approach with improved training recipes. However, latent-space objectives risk abstract drift: the representations are not anchored to observations (images or sentences), inducing lossy compression and discarding cues (*e.g.* color) critical for dense prediction. In Sapiens2, we combine the image-anchored MAE objective with the semantic CL objective. Prior work such as CMAE (Huang et al., 2023) explores this combination but evaluates primarily on classification. In contrast, we study a unified objective at billion-scale across multiple human-centric tasks.

**Human-Centric Vision Models.** Many recent works focus on building models for human-centric vision. These models often outperform general models of similar scale on human-related tasks. For instance, HAP (Yuan et al., 2023) uses 2D keypoints to guide the mask sampling process during masked image modeling, encouraging the model to focus on body structure information. Geoman (Kim et al., 2025a) uses an image-to-video diffusion model for geometry estimation. HCMoCo (Hong et al., 2022) and PBoP (Meng et al., 2024) employ multiple encoders to exploit multimodal human body consistency through a hierarchical contrastive learning framework. SOLIDER (Chen et al., 2023) introduces a human semantic classification loss to inject semantic

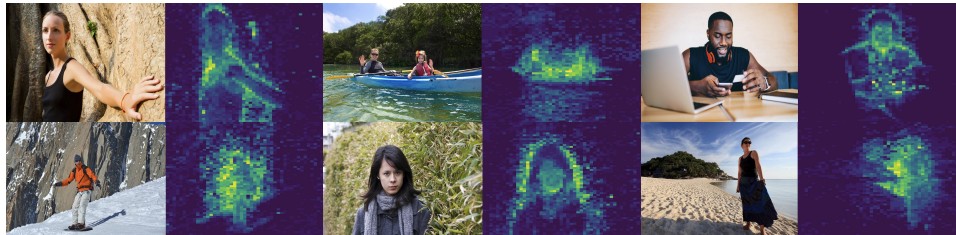

Figure 3: **Human-centric attention.** Visualization of [CLS]-token self-attention across heads in the final layer.

information into the learned features. LiftedCL (Chen et al., 2022) incorporates an adversarial loss to supervise the lifted 3D skeletons, explicitly embedding 3D human structure information for human-centric pretraining. SapiensID (Kim et al., 2025b) trains a model specifically for person re-identification. In contrast to these approaches, Sapiens2 does not inject any explicit human priors beyond the data itself during pretraining. This truly inductive prior-free approach enables scaling to millions of images and model sizes without introducing handcrafted human-centric biases.

**Vision Transformers at Scale.** Although the largest vision backbones remain an order of magnitude smaller than language models (Lu et al., 2024), the field is scaling rapidly as both data and model sizes grow. To clarify the landscape, we position prior works along three axes: parameters, resolution, and data. Amongst notable recent works, the largest vision backbone in the Perception Encoder family (Bolya et al., 2025) has 2B parameters, is trained at 448 px resolution, and uses 5.4B samples. DINOv2 (Oquab et al., 2023) scales to 1B parameters at 512 px and is pretrained on 152M images. ViT-22B (Dehghani et al., 2023) remains the largest model by parameter count; it is trained at 224 px and is pretrained on 1M images from ImageNet (Russakovsky et al., 2015). Sapiens-2B (Khirodkar et al., 2024), at 1024 px, was the largest human-centric vision backbone, pretrained on 300M human images. In Sapiens2, we scale to 5B parameters and extend the input resolution to 4K, yielding a vision backbone with the largest FLOPs, trained on 1B human images.

## 3 PRETRAINING

This section details our pretraining data and methodology, with emphasis on human-centric curation and design choices that preserve output fidelity and strengthen semantic understanding.

### 3.1 HUMANS-1B DATASET

Scale helps only when the data distribution is diverse, balanced, and high quality (Touvron et al., 2023; Radford et al., 2021; Chuang et al., 2025). From a web-scale pool of $\sim$4B images, we isolate human-centric content via a multi-stage filter: bounding box detection, head-pose estimation, aesthetic and realism scoring, CLIP (Radford et al., 2021) features and text-overlay detection. We remove images that fail realism, quality or other checks. From the remainder, we retain instances where at least one person is $\geq 384$ pixels on the short side; images may contain multiple people. We deduplicate via perceptual hashing and deep-feature nearest-neighbor pruning, and we cluster visual embeddings followed by selective sampling (Oquab et al., 2023) to balance content across poses, viewpoints, occlusion, clothing, scene types, and illumination. Thresholds and balance caps are calibrated with small human audits. The result is a curated, balanced corpus of $\sim$1B high-quality human images for pretraining.

### 3.2 SELF-SUPERVISED LEARNING

Let $\mathcal{I}$ denote the training set. We sample an image $\mathbf{x} \sim \mathcal{I}$ and draw $V$ random augmentations to obtain views $\{\mathbf{x}_i\}_{i=1}^{V}$. Each view is patchified into $N$ tokens indexed by $\mathcal{P} = \{1, \dots, N\}$, i.e., $\mathbf{x}_i = \{\mathbf{x}_i^p\}_{p \in \mathcal{P}}$. Let $\{\mathbf{e}_{\text{pos}}^p\}_{p \in \mathcal{P}}$ be positional embeddings (Dosovitskiy et al., 2020) and $\Phi_{\text{enc}}$, $\Phi_{\text{dec}}$, $\Phi_{\text{cls}}$ be our transformer encoder, patch decoder and contrastive decoder respectively. Specifically, $\Phi_{\text{cls}}$ maps the encoder [CLS] token to $K$ logits.

**Masked Image Modeling.** For each view $i \in \{1, \dots, V\}$, we sample a binary mask $\mathbf{m}_i \in \{0,1\}^N$ with masking ratio $r$. The masked and visible index sets are defined as $\mathcal{M}_i = \{p \in \mathcal{P} : m_i^p = 1\}$ and $\mathcal{V}_i = \mathcal{P} \setminus \mathcal{M}_i$. The encoder $\Phi_{\text{enc}}$ processes only visible tokens: $\mathbf{z}_i^{\text{vis}} = \Phi_{\text{enc}}(\{\mathbf{x}_i^p + \mathbf{e}_{\text{pos}}^p\}_{p \in \mathcal{V}_i})$. We then form a full sequence by scattering $\mathbf{z}_i^{\text{vis}}$ back to $\mathcal{V}_i$ and inserting a learned mask token at $\mathcal{M}_i$: $\mathbf{z}_i = \text{scatter}(\mathbf{z}_i^{\text{vis}}; \mathcal{V}_i) \cup \{\mathbf{e}_{[\text{MASK}]} + \mathbf{e}_{\text{pos}}^p\}_{p \in \mathcal{M}_i}$. The decoder $\Phi_{\text{dec}}$ reconstructs all patches, $\hat{\mathbf{x}}_i = \Phi_{\text{dec}}(\mathbf{z}_i)$ with outputs $\{\hat{\mathbf{x}}_i^p\}_{p \in \mathcal{P}}$. Following He et al. (2022), targets are normalized $\tilde{\mathbf{x}}_i^p$, and

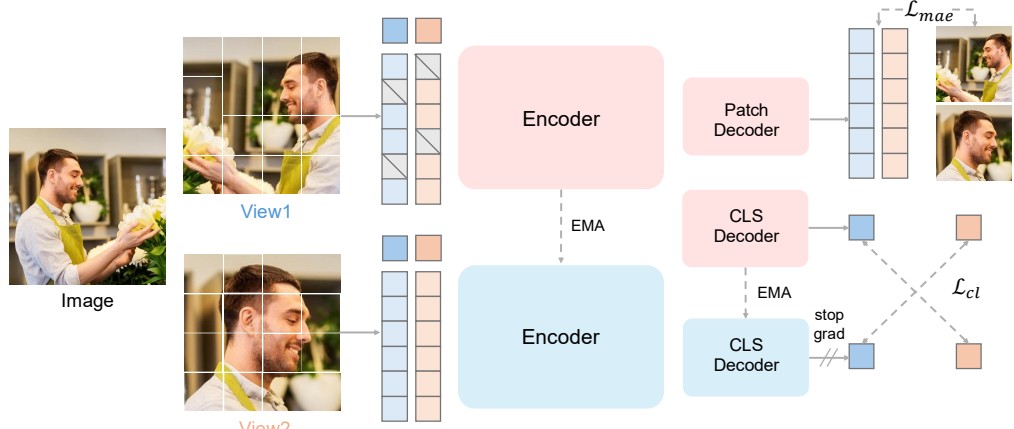

Figure 4: **SAPIENS2 Pretraining**. We combine the masked reconstruction loss ($\mathcal{L}_{\mathrm{mae}}$) with a global contrastive loss on [CLS] ($\mathcal{L}_{\mathrm{cl}}$). Multiple image views are generated, and a student–teacher framework matches predicted distributions across views. $\mathcal{L}_{\mathrm{mae}}$ helps the model learn low-level details (*e.g.*texture) for high-fidelity dense tasks, while $\mathcal{L}_{\mathrm{cl}}$ improves semantic understanding across human images.

the loss averages MSE over *masked* tokens and views:

$$\mathcal{L}_{\mathrm{MAE}} = \frac{1}{V} \sum_{i=1}^{V} \frac{1}{|\mathcal{M}_i|} \sum_{p \in \mathcal{M}_i} \left\| \tilde{\mathbf{x}}_i^p - \hat{\mathbf{x}}_i^p \right\|_2.$$

**Contrastive Learning.** We adopt a student–teacher scheme based on DINOv3 (Siméoni et al., 2025); the teacher has the same architecture ($\Phi_{\mathrm{enc}}, \Phi_{\mathrm{cls}}$), is *non-learnable*, and its parameters are an EMA of the student. For each view $i$, the student and teacher [CLS] embeddings and logits are

$$\mathbf{c}_i^s = [\text{CLS}](\Phi_{\mathrm{enc}}(\mathbf{x}_i)), \quad \mathbf{c}_i^t = [\text{CLS}](\Phi_{\mathrm{enc}}^{\mathrm{ema}}(\mathbf{x}_i)), \qquad \mathbf{s}_i = \Phi_{\mathrm{cls}}(\mathbf{c}_i^s), \quad \mathbf{t}_i = \Phi_{\mathrm{cls}}^{\mathrm{ema}}(\mathbf{c}_i^t),$$

with $\mathbf{p}_i = \mathrm{softmax}(\mathbf{s}_i)$ and $\mathbf{q}_i = \mathrm{softmax}(\mathbf{t}_i)$. For the $V$-view (global + local) setting, we form the positive pair set $\mathcal{S}$ consisting of all cross-view global↔global and global↔local pairs (excluding same-view matches for global crops; local↔local pairs are skipped). The contrastive objective averages a teacher-to-student cross-entropy over these pairs:

$$\mathcal{L}_{\mathrm{CL}} = \frac{1}{|\mathcal{S}|} \sum_{(i,j) \in \mathcal{S}} H(\mathbf{q}_j, \mathbf{p}_i), \qquad H(\mathbf{q}, \mathbf{p}) = -\sum_{k=1}^{K} q_k \log p_k.$$

Finally, Fig. 4 shows our pretraining setup for $V = 2$; for clarity, the figure depicts the global contrastive objective only. We use a joint objective $\mathcal{L} = \mathcal{L}_{\mathrm{MAE}} + \lambda \mathcal{L}_{\mathrm{CL}}$, combining human-centric low-level fidelity with view-invariant semantics.

## 4 MODEL ARCHITECTURE

We revise the backbone to stably scale to 5B parameters, increase the input resolution from 1K to 4K, and maintain compatibility with sparse masked pretraining. The mid-depth blocks use grouped-query attention (GQA) (Ainslie et al., 2023), while the early and late blocks use standard multi-head self-attention. We replace the feed-forward layers with gated SwiGLU-FFN variants (Shazeer, 2020). For long-schedule stability, we apply QK-Norm (Henry et al., 2020)—normalizing queries and keys before attention—and substitute LayerNorm with the

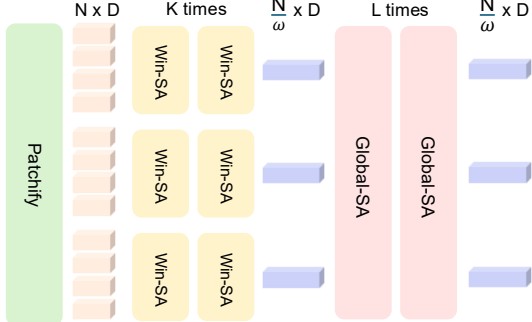

Figure 5: **Windowed self-attention** for 4K resolution.

parameter-efficient RMSNorm (Zhang & Sennrich, 2019). To scale to 4K inputs, we adopt a hierarchical attention design (Ryali et al., 2023): given an $H \times W$ image with patch size $p$, yielding $N = (H/p)(W/p)$ tokens, the first $K$ layers apply windowed self-attention to capture local structure. We then downsample the 2D token grid by a spatial stride $\sqrt{\omega}$ via [CLS]-guided pooling to obtain $N/\omega$ tokens. Next $L$ layers use global attention over this reduced sequence, refer Fig. 5. During pretraining, we apply token masking after the local stage, and include a brief masked-reconstruction phase at 2K to sharpen sub-pixel fidelity on dense tasks without degrading semantics. Finally, we increase decoder outputs to 1K for base backbones (from 0.5K) and to 2K for 4K backbones.

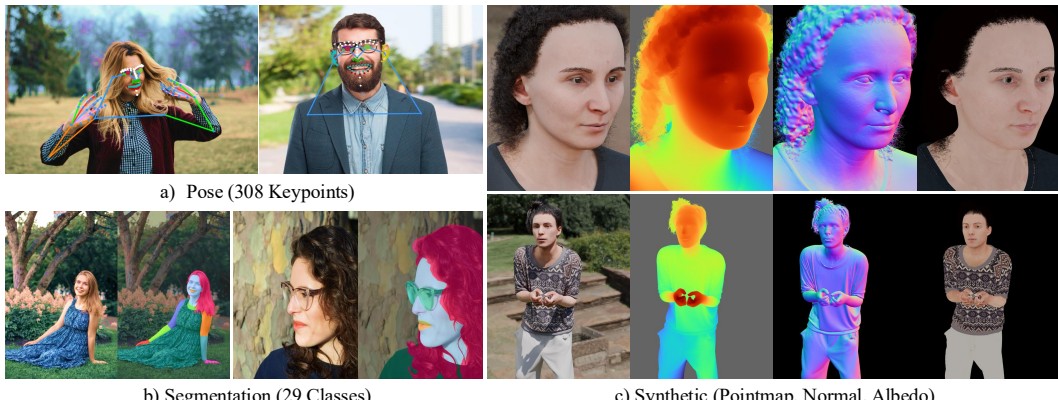

a) Pose (308 Keypoints)

b) Segmentation (29 Classes)

c) Synthetic (Pointmap, Normal, Albedo)

Figure 6: **Post-Training Annotations.** We annotated 100K in-the-wild images with *pose (a)* and *segmentation (b)*, class vocabulary is also extended to include eyeglasses (in cyan). For *pointmap, normal, albedo (c)*, we improve our synthetic assets to capture finer geometric details and color variations.

## 5 POST-TRAINING

We fine-tune the pretrained backbone on five human-centric tasks—pose estimation, body-part segmentation, depth, surface normals, and albedo—using lightweight task-specific heads while leaving the backbone unchanged. Relative to Khirodkar et al. (2024), we broaden supervision and refine task objectives.

**Pose Estimation.** We follow a top-down paradigm to estimate keypoint heatmaps from an input image. Our keypoint topology is a 308-keypoint full-body skeleton with dense coverage of the face (243) and hands (40 total), with the remainder spanning torso and lower-body. Unlike Khirodkar et al. (2024), which relied solely on capture-studio annotations, we add in-the-wild supervision (Fig. 6a) by newly annotating 100K high-resolution images from our pretraining corpus with the same vocabulary. This hybrid supervision improves generalization to unconstrained images. Our objective uses MSE over ground-truth heatmaps with OHEM (Chen et al., 2018) to focus supervision within a large keypoint set as $\mathcal{L}_{\text{pose}} = \sum_{u \in \Omega} \|\hat{\mathbf{H}}(u) - \mathbf{H}(u)\|_2$.

**Body-Part Segmentation.** Our segmentation vocabulary has 29 classes (extended from the previous 28 by adding eyeglasses; see Fig. 6b). The vocabulary targets part-specific supervision and precise localization of semantic human body parts. Similar to pose, we increase segmentation supervision to 20K in-the-wild images with segmentation labels. Our objective uses per-pixel weighted cross-entropy combined with Dice loss (Azad et al., 2023) for sharper boundaries.

**Pointmap (Depth) Estimation.** Rather than relative depth, we regress a per-pixel 3D pointmap $\hat{\mathbf{P}}(u) \in \mathbb{R}^3$ in the camera frame. Since metric scale is ambiguous with unknown intrinsics (Yin et al., 2023), we predict a focal-normalized pointmap $\tilde{\mathbf{P}}(u)$ and a scalar head $s$, forming $\hat{\mathbf{P}}(u) = s\tilde{\mathbf{P}}(u)$ (Bochkovskii et al., 2024). Supervision is entirely synthetic and uses higher-fidelity assets (hair, eyes, fine facial wrinkles, Fig. 6c). The loss is $\mathcal{L}_{\text{pointmap}} = \sum_{u \in \Omega} \|\hat{\mathbf{P}}(u) - \mathbf{P}(u)\|_2 + \|\nabla \hat{\mathbf{P}}(u) - \nabla \mathbf{P}(u)\|_2$ where $\nabla$ is finite differences along XY.

**Normal Estimation.** We predict per-pixel unit normals $\hat{\mathbf{N}}(u) \in \mathbb{R}^3$ for human pixels using the same high-fidelity synthetic assets; the decoder uses multiple PixelShuffle (Aitken et al., 2017) layers for artifact-free upsampling. The loss is defined as: $\mathcal{L}_{\text{normal}} = \sum_{u \in \Omega}(1 - \hat{\mathbf{N}}(u) \cdot \mathbf{N}(u)) + \|\hat{\mathbf{N}}(u) - \mathbf{N}(u)\|_2 + \|\nabla \hat{\mathbf{N}}(u) - \nabla \mathbf{N}(u)\|_2$.

**Albedo Estimation.** We predict per-pixel diffuse albedo $\hat{\mathbf{A}}(u) \in [0, 1]^3$, crucial for relighting (Kim et al., 2024). Training uses high-fidelity synthetic pairs $\mathbf{A}(u)$ (Fig. 6c) and encourages illumination-invariant recovery of skin tone and clothing. The loss is $\mathcal{L}_{\text{albedo}} = \sum_{u \in \Omega} \|\hat{\mathbf{A}}(u) - \mathbf{A}(u)\|_2 + \|\nabla \hat{\mathbf{A}}(u) - \nabla \mathbf{A}(u)\|_2 + \|\mu(\hat{\mathbf{A}}) - \mu(\mathbf{A})\|_2$, where $\mu(\cdot)$ is the spatial RGB mean for alignment.

## 6 EXPERIMENTS

In this section, we initially outline implementation details, then evaluate pretrained feature generalization using dense probing and post-train performance across a variety of downstream tasks.

| Model | Parent-Model | #Params | FLOPs | Hidden size | Layers | Heads |
|---|---|---|---|---|---|---|
| Sapiens2-0.4B | Sapiens-0.3B | 0.398 B | 1.260 T | 1024 | 24 | 16 |
| Sapiens2-0.8B | Sapiens-0.6B | 0.818 B | 2.592 T | 1280 | 32 | 16 |
| Sapiens2-1B | Sapiens-1B | 1.462 B | 4.715 T | 1536 | 40 | 24 |
| Sapiens2-5B | - | 5.071 B | 15.722 T | 2432 | 56 | 32 |

Table 1: **SAPIENS2 architectural details**. Broadly, we base the smaller models on the first generation and introduce a 5B variant that scales both depth (layers) and width (token embeddings).

## 6.1 IMPLEMENTATION DETAILS

Sapiens2 is implemented in PyTorch with HF-Accelerate (Gugger et al., 2022). All our models are trained on A100 GPUs using bfloat16 and FSDP for efficiency. We use fused AdamW (Loshchilov & Hutter, 2017) as the optimizer for all experiments, with a brief learning-rate warmup followed by cosine decay. We pretrain from scratch at $1024 \times 768$ (1K) and $4096 \times 3072$ (4K) resolutions. Starting from Sapiens–0.3B, 0.6B and 1B, we apply the architectural revisions in Sec. 2 to produce Sapiens2–0.4B, 0.8B and 1B. To push the frontier for human-centric vision models, we also introduce a 5B model that scales both network depth and token embedding dimensions. Sapiens2-5B is the highest-FLOPs vision transformer at 15 TFlops. Table 1 summarizes our model configurations at 1K resolution. Finally, we fine-tune the 1B–4K model for segmentation and normal estimation.

**Evaluation**. We construct task-specific test sets to measure fidelity and generalization, and importantly go beyond existing benchmarks in annotation quality. Each set contains challenging in-the-wild samples. For pose, we evaluate on 11K images annotated with 308 keypoints, in contrast to the 5K capture-studio images used by SAPIENS. For segmentation, we use a similar in-the-wild test of 5K images with 29 classes. For pointmap, normals, and albedo, following Saleh et al. (2025), we evaluate on a 10K-image test set built from our photorealistic assets with higher geometric detail. Please refer to the appendix for additional details.

## 6.2 PRETRAINING GENERALIZATION: DENSE PROBING

To evaluate zero-shot generalization of the pretrained backbone, we perform dense probing and compare against state-of-the-art vision backbones—Sapiens (Khirodkar et al., 2024), PE (Bolya et al., 2025), DINOv2 (Oquab et al., 2023), and DINOv3 (Siméoni et al., 2025)—across a variety of human tasks. For dense probing, we freeze the backbone and lightly train a task-specific decoder with identical hyperparameters across all methods. The tasks vary in their demands: for pose estimation, high-level human semantics aid keypoint localization, whereas for albedo recovery, the backbone must closely capture input appearance. Table 2 reports task-specific metrics across multiple model sizes. Among baselines, DINOv3 is strongest for pose and geometric understanding (e.g., pointmaps), owing to its contrastive objective and scale. Sapiens (Khirodkar et al., 2024),

| Model | Params | Pose | | Seg | | Pointmap | Normal | | Albedo |
|---|---|---|---|---|---|---|---|---|---|
| | | mAP ↑ | mAR ↑ | mIoU (%) ↑ | mAcc (%) ↑ | L2 ↓ | MAE° ↓ | % 22.5° ↑ | MAE ($\times 10^{-2}$) ↓ |
| PE-L (Bolya et al., 2025) | 0.30B | 34.8 | 38.4 | 42.1 | 62.3 | 0.537 | 17.9 | 74.5 | 4.22 |
| PE-H (Bolya et al., 2025) | 0.63B | 50.2 | 53.8 | 45.8 | 65.3 | 0.529 | 17.1 | 76.2 | 4.14 |
| DINOv2-G (Oquab et al., 2023) | 1.14B | 59.5 | 63.1 | 62.7 | 78.9 | 0.432 | 15.0 | 80.7 | 3.92 |
| Sapiens-1B (Khirodkar et al., 2024) | 1.17B | 58.2 | 61.8 | 61.4 | 78.2 | 0.532 | 15.3 | 80.1 | 3.85 |
| Sapiens-2B (Khirodkar et al., 2024) | 2.16B | 63.4 | 66.9 | 65.1 | 80.6 | 0.515 | 14.6 | 81.4 | 3.72 |
| DINOv3-B (Siméoni et al., 2025) | 0.11B | 51.7 | 55.3 | 62.6 | 78.9 | 0.492 | 16.2 | 78.0 | 4.08 |
| DINOv3-L (Siméoni et al., 2025) | 0.34B | 63.8 | 66.8 | 65.5 | 80.0 | 0.465 | 15.6 | 79.7 | 3.95 |
| DINOv3-H (Siméoni et al., 2025) | 0.88B | 67.6 | 70.4 | 65.4 | 81.4 | 0.448 | 15.2 | 80.5 | 3.86 |
| DINOv3-7B (Siméoni et al., 2025) | 6.71B | 68.2 | 71.6 | 67.6 | 83.3 | 0.398 | 14.2 | 82.5 | 3.48 |
| Sapiens2-0.4B (Ours) | 0.39B | 65.2 | 68.2 | 64.8 | 79.9 | 0.471 | 15.0 | 80.5 | 3.96 |
| Sapiens2-0.8B (Ours) | 0.82B | 66.2 | 69.1 | 66.9 | 81.8 | 0.435 | 14.4 | 81.9 | 3.89 |
| Sapiens2-1B (Ours) | 1.46B | 68.3 | 71.4 | 65.2 | 82.9 | 0.428 | 14.5 | 81.6 | 3.64 |
| Sapiens2-5B (Ours) | 5.07B | **74.7** (+6.5) | **77.4** (+5.8) | **69.6** (+2.0) | **83.5** (+0.2) | **0.358** (-0.04) | **13.5** (-0.7) | **83.7** (+1.2) | **3.12** (-0.36) |

Table 2: **Dense probing on human tasks.** We freeze the backbone and fine-tune a lightweight, task-specific decoder with identical hyperparameters across all methods.

| Model | Input Size | mAP (%) | mAR (%) |
|---|---|---|---|
| ViTPose+-L, TPAMI23 | $256 \times 192$ | 47.8 | 53.6 |
| ViTPose+-H, TPAMI23 | $256 \times 192$ | 48.3 | 54.1 |
| DWPose-M, ICCV23 | $256 \times 192$ | 60.6 | 67.4 |
| DWPose-L, ICCV23 | $384 \times 288$ | 66.5 | 72.8 |
| RTMW-L, arxiv23 | $384 \times 288$ | 70.1 | 75.9 |
| RTMW-X, arxiv23 | $384 \times 288$ | 70.2 | 76.1 |
| Sapiens-1B*, ECCV24 | $1024 \times 768$ | 76.8 | 79.3 |
| Sapiens-2B*, ECCV24 | $1024 \times 768$ | 78.3 | 82.1 |
| Sapiens2-0.4B (Ours) | $1024 \times 768$ | 76.9 | 81.3 |
| Sapiens2-0.8B (Ours) | $1024 \times 768$ | 79.4 (+1.1) | 83.1 (+1.0) |
| Sapiens2-1B (Ours) | $1024 \times 768$ | 80.4 (+2.1) | 84.0 (+1.9) |
| Sapiens2-5B (Ours) | $1024 \times 768$ | **82.3** (+4.0) | **85.3** (+3.2) |

Table 3: **Pose estimation** on 11K `test`. Flip test is used, same detections. *Denotes *v1* open-sourced models.

| Model | mIoU (%) | mAcc (%) |
|---|---|---|
| SegFormer. Neurips21 | 45.2 | 68.3 |
| Mask2Former, CVPR22 | 48.7 | 71.5 |
| DeepLabV3+, ECCV18 | 42.8 | 66.9 |
| HRNetV2+OCR | 47.3 | 70.2 |
| Sapiens-1B*, ECCV24 | 53.8 | 74.7 |
| Sapiens-2B*, ECCV24 | 58.2 | 77.2 |
| Sapiens2-0.4B (Ours) | 79.5 (+21.3) | 90.9 (+13.7) |
| Sapiens2-0.8B (Ours) | 80.6 (+22.4) | 90.2 (+13.0) |
| Sapiens2-1B (Ours) | 81.7 (+23.5) | 91.6 (+14.4) |
| Sapiens2-1B-4K (Ours) | 81.9 (+23.7) | **92.0** (+14.8) |
| Sapiens2-5B (Ours) | **82.5** (+24.3) | 91.1 (+13.9) |

Table 4: **Segmentation** on 5K `test`. All methods have the same `train` set. *Denotes *v1* open-sourced models.

due to its masked-autoencoder pretraining, has limited semantic understanding but retains low-level appearance cues useful for albedo estimation. With our combined pretraining objective, Sapiens2 outperforms baselines at comparable model sizes, and our largest model, Sapiens2-5B, surpasses all baselines across every task.

## 6.3 COMPARISON WITH STATE-OF-THE-ART METHODS

To understand performance and generalization across human-centric tasks, we compare our models against task-specific state-of-the-art methods in this section. We provide a brief summary here and refer to the appendix for detailed analysis.

**Pose.** We compare Sapiens2 with state-of-the-art whole-body top-down pose estimators in Table 3. We retrain baselines on our new keypoint set using recommended settings. Our models substantially improve over the first generation; specifically, Sapiens2-0.8B, despite its smaller parameter count, outperforms larger models due to architectural improvements and broader supervision. Consistent with scaling laws Kaplan et al. (2020), our results show predictable gains with increased scale. Our largest model, Sapiens2-5B, sets a new state of the art for dense 308-keypoint predictions in-the-wild, achieving 82.3 mAP on challenging poses.

**Segmentation.** Table 4 compares our models to state-of-the-art methods on our segmentation vocabulary. For fairness, we train all baselines on our training set. Sapiens2 generalizes strongly to in-the-wild images with high-resolution outputs. Although the input resolution is the same (1K) for Sapiens and Sapiens2, Sapiens2–1B outperforms Sapiens-1B by 27.9% mIoU and 16.9% mAcc, owing to in-the-wild supervision and an increased output resolution of 1K (from 0.5K).

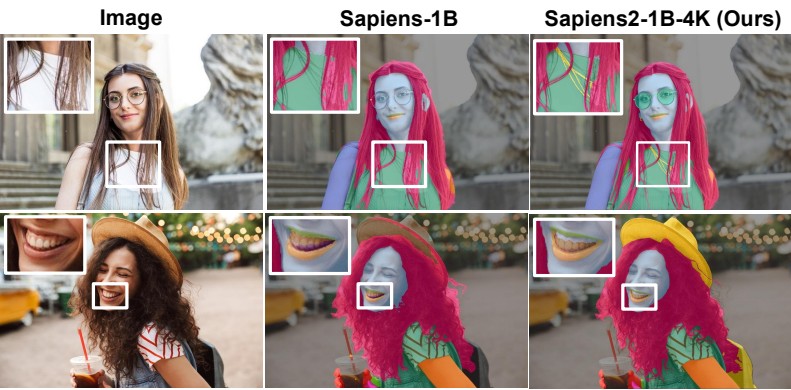

Figure 7: **Body-part segmentation** using our 1B-4K model.

| Method | Distance | | Abs. Error | | |
|---|---|---|---|---|---|
| | L2 ($e^{-1}$) | RMSE | X ($e^{-3}$) | Y ($e^{-3}$) | Z ($e^{-2}$) |
| UniDepth, CVPR24 | 0.368 | 0.689 | 8.34 | 10.92 | 5.23 |
| DUSt3R, CVPR24 | 0.349 | 0.663 | 7.66 | 10.11 | 4.86 |
| VGGT, CVPR25 | 0.217 | 0.515 | 3.79 | 4.96 | 2.19 |
| MoGe, CVPR25 | 0.202 | 0.486 | 3.21 | 4.41 | 1.89 |
| Sapiens2-0.4B (Ours) | 0.190 | 0.466 | 3.15 | 4.33 | 1.76 |
| Sapiens2-0.8B (Ours) | 0.186 | 0.459 | 3.12 | 4.26 | 1.72 |
| Sapiens2-1B (Ours) | 0.178 | 0.478 | 2.95 | 4.01 | 1.66 |
| Sapiens2-5B (Ours) | **0.167** | **0.443** | **2.85** | **3.86** | **1.55** |

Table 5: **Pointmap evaluation** in focal-length normalized canonical coordinates on 10K `test`.

**Pointmap.** Table 5 compares Sapiens2 with existing pointmap (XYZ) estimation methods such as UniDepth (Piccinelli et al., 2024), DUSt3R (Wang et al., 2024), VGGT (Wang et al., 2025a), and MoGe Wang et al. (2025b). This task is more challenging than relative depth estimation, as it requires reasoning about camera intrinsics. For fairness, we optimize for scale and evaluate all predictions in a focal-length-normalized canonical space. Our models outperform all baselines, including MoGe (Wang et al., 2025b), across all model sizes. Fig. 8 qualitatively compares Sapiens2-1B with MoGe, showing that our predicted pointmaps better preserve human-specific geometric details.

**Normal.** We compare our finetuned normal estimators with current state-of-the-art monocular methods in Table 6. Our evaluation set consists of whole-body scan images captured from random virtual camera viewpoints, with ground-truth normals available at $4K$ resolution. Our smallest model, Sapiens2-0.4B, outperforms existing methods by achieving a mean angular error of $8.63°$, with $94.76\%$ of human pixels below the $30°$ threshold. Fig. 9 compares Sapiens2 with the baseline DAViD Saleh et al. (2025) and shows that it captures geometric details accurately and remains robust under varying lighting conditions.

**Albedo.** Table 7 reports quantitative albedo results on our 10K `test` set. Our models show consistent improvement with scale; Sapiens2-5B achieves the lowest MAE of $0.012$ and highest PSNR of $32.6$ dB. Despite training solely on synthetic data, our model recovers true skin tone under varying

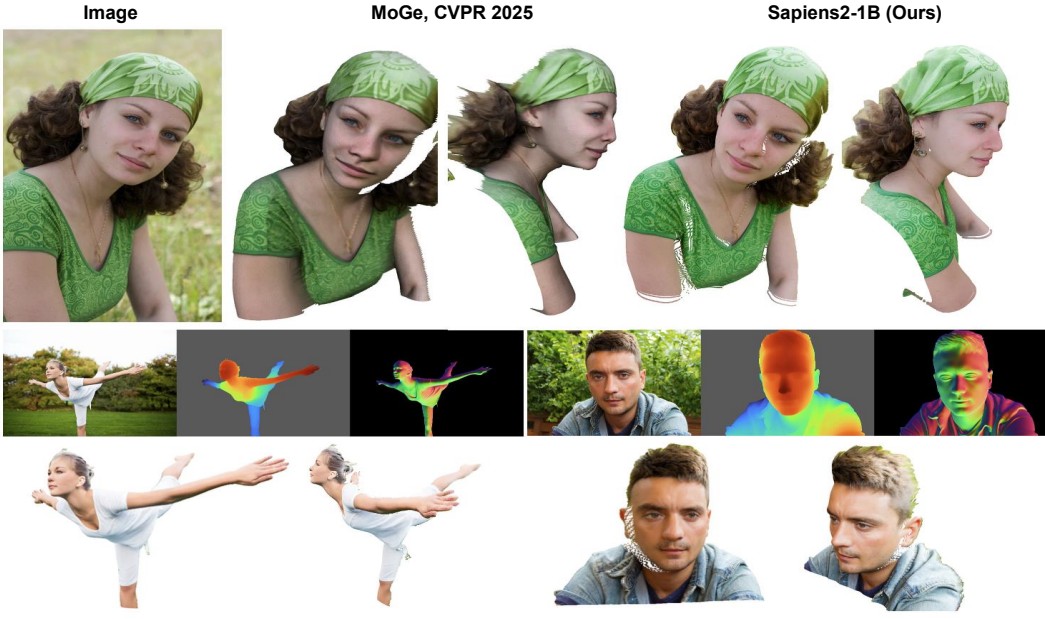

Figure 8: (*Top*) **Pointmap** qualitative comparison of Sapiens2-1B with MoGe (Wang et al., 2025b). (*Bottom*) Depth visualized from the predicted pointmap, along with surface normals and novel 3D viewpoints.

| Method | Angular Error° | | % Within $t°$ | | |
|---|---|---|---|---|---|
| | Mean | Median | 5° | 11.25° | 30° |
| Marigold, CVPR24 | 18.83 | 15.27 | 9.41 | 39.87 | 45.21 |
| DSINE, CVPR24 | 17.24 | 13.51 | 11.67 | 45.62 | 48.79 |
| Sapiens-1B* ECCV24 | 13.62 | 10.11 | 32.18 | 69.34 | 82.14 |
| Sapiens-2B* ECCV24 | 12.38 | 9.46 | 37.05 | 70.54 | 85.62 |
| DAViD-L, ICCV25 | 10.73 | 7.49 | 42.91 | 72.16 | 89.27 |
| Sapiens2-0.4B (Ours) | 8.63 | 5.25 | 49.13 | 76.89 | 94.76 |
| Sapiens2-0.8B (Ours) | 8.49 | 4.75 | 51.18 | 77.19 | 94.81 |
| Sapiens2-1B (Ours) | 7.12 | 3.75 | 58.31 | 81.69 | 95.77 |
| Sapiens2-1B-4K (Ours) | 6.98 | 3.08 | 59.07 | 82.10 | 95.88 |
| Sapiens2-5B (Ours) | **6.73** | **2.74** | **62.80** | **83.06** | **96.13** |

Table 6: **Normal evaluations** on 10K whole-body `test` set at 4K ground-truth resolution.

| Model | MAE | RMSE | PSNR | SSIM | Grad-L1 |
|---|---|---|---|---|---|
| Sapiens2-0.4B | 0.01825 | 0.03257 | 29.74 | 0.889 | 0.00642 |
| Sapiens2-0.8B | 0.01602 | 0.02876 | 30.83 | 0.903 | 0.00624 |
| Sapiens2-1B | 0.01224 | 0.02392 | 32.43 | 0.914 | 0.00612 |
| Sapiens2-5B | **0.01191** | **0.02341** | **32.61** | **0.915** | **0.00610** |

Table 7: **Albedo estimation** on 10K `test` set with ground-truth from synthetic renders.

lighting conditions and generalizes to in-the-wild images (Fig. 10). Unlike diffusion-based methods Liang et al. (2025), our model is feedforward and significantly more efficient at inference.

## 7 CONCLUSION

SAPIENS2 introduces high-resolution, human-centric models pretrained on a 1-billion-image dataset. Our models simultaneously learn appearance cues and semantics by combining masked reconstruction and contrastive objectives. They consistently outperform general-purpose models on human images and extend to tasks ranging from pose estimation to albedo recovery. SAPIENS2 sets a new benchmark for high-fidelity dense predictions and provides a robust foundation for applications requiring a nuanced, detailed understanding of humans in unconstrained visual contexts.

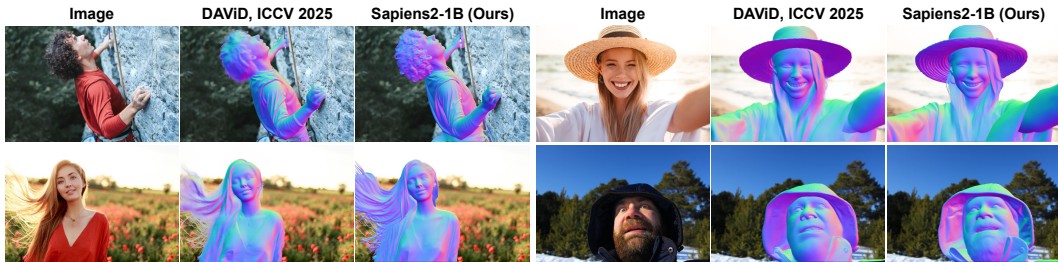

Figure 9: **Normal prediction**. Qualitative comparison of Sapiens2-1B with DAViD (Saleh et al., 2025).

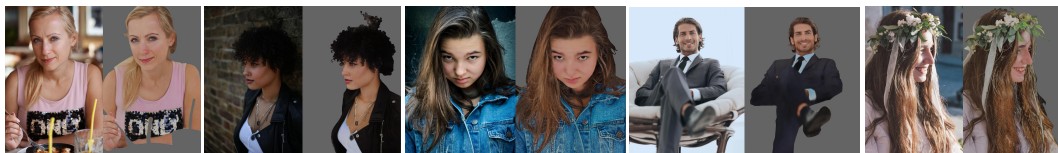

Figure 10: **Albedo estimation** using Sapiens2-1B. Our model effectively encodes low-level details crucial for albedo estimation and generalizes well to in-the-wild images, despite being trained on limited synthetic data.

## ACKNOWLEDGMENTS

We gratefully acknowledge the following individuals for their contributions and support: Amaury Aubel, Sofien Bouaziz, Nicholas Dahm, Simon Dong, Lucas Evans, Ish Habib, Kris Kitani, Devansh Kukreja, Junxuan Li, Maxime Oquab, Tero Pikkarainen, Don Pinkus, Kaila Prochaska, Wei Pu, Nir Sopher, Jess Wiese.

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

# A   APPENDIX

## A.1   PRETRAINING

### A.1.1   IMPLEMENTATION DETAILS

We use the dense-probing evaluations as the final metrics to guide any design decisions during the pretraining stage. For instance, we pretrain the SAPIENS2–1B (embed dim 1536, 40 layers, 24 heads, patch size 16, final norm with [CLS]) at $1024{\times}768$. Training uses a joint MAE and contrastive objective: an 8-layer MAE decoder (dim 512) with $\ell_2$ reconstruction, and a [CLS] projection head for contrastive learning. Loss weights are MAE: 1.0, CLS: 0.4, KoLeo: 0.04. We adopt multi-view training with 2 global and 4 local crops; global crops use random resize–crop in ratio $[0.5, 1.0]$, local crops in $[0.2, 0.7]$, with standard color/blur/solarize and horizontal flips. Inputs are normalized to ImageNet means/stds. Importantly, we do not use color augmentations on the global views - used for masked reconstruction objective.

Optimization uses fused AdamW (lr $1{\times}10^{-4}$, $(\beta_1, \beta_2){=}(0.9, 0.95)$, wd 0.05) with zero-decay for norms, biases, positional and special tokens. We train for $5{\times}10^5$ iters with $10^3$ warmup, cosine decay to $10^{-7}$, and global grad-norm clip 5.0. The contrastive teacher EMA is 0.992 (center momentum 0.9); student temperature is 0.1, teacher temperature warms from 0.065 to 0.07 over the first $10^3$ iters. We evaluate every checkpoint for downstream tasks with a frozen encoder and report results using the best checkpoint.

### A.1.2   MASKING STRATEGY

Given the high resolution of our backbones, we use mixed blockwise/patchwise masking (blockwise prob 0.4) with a 75% mask ratio at patch size 16, refer Fig. 11. At $1024{\times}768$ ($64{\times}48{=}3072$ patches), this masks $\sim 2304$ patches per image, yielding coarse occlusions that regularize MAE while leaving sufficient context for contrastive learning.

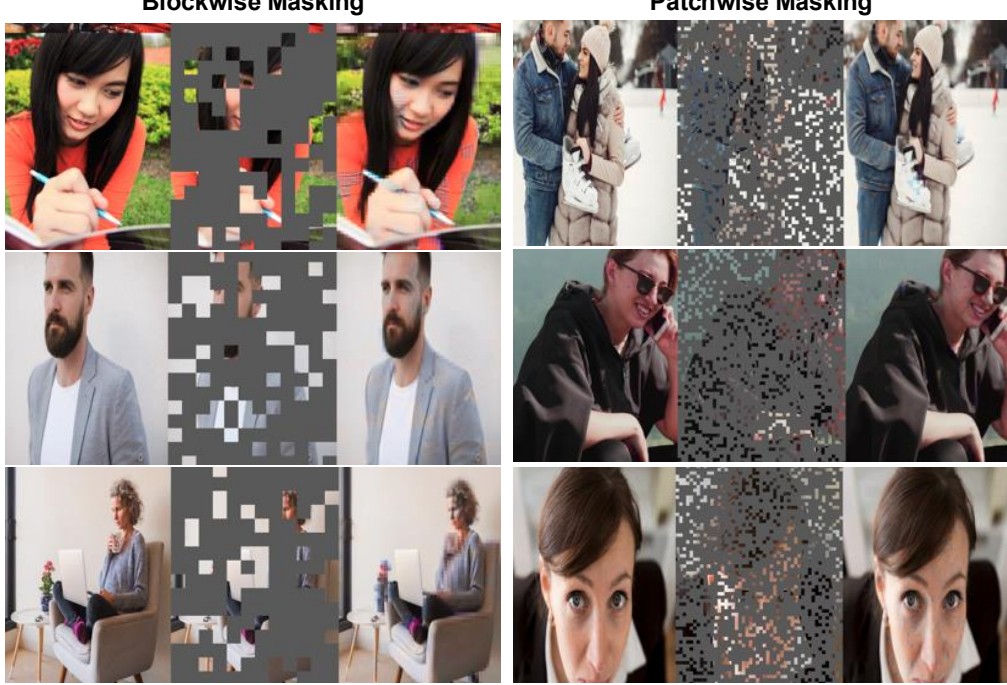

Figure 11: We randomly mix blockwise and patchwise masking to provide coarse occlusions. For MAE pretraining at high resolution (1024), we use a 75% mask ratio. Each sample represents (ground-truth image, masked input, reconstruction).

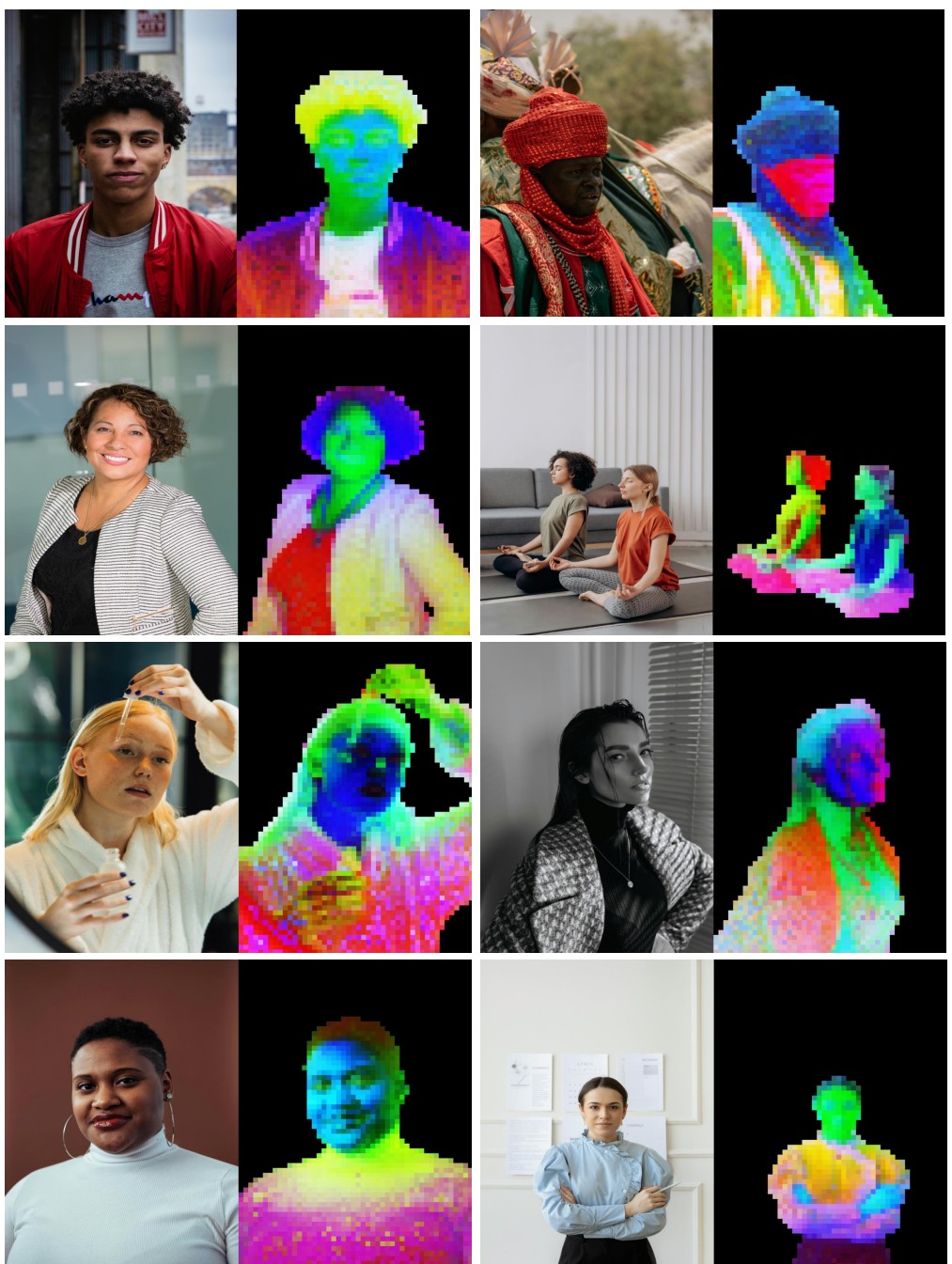

Figure 12: We visualize the encoder features using PCA (3 major components) with different colors. We use foreground masking to extract patch features for human pixels. Sapiens2 features capture texture and color information as well as showcase human semantics.

## A.2 POSE ESTIMATION

We evaluate Sapiens2 using ground-truth bounding boxes on our in-the-wild `test` set for 308 keypoints. We fine-tune a top-down pose estimator initialized from a pretrained checkpoint with the `[CLS]` token disabled so the encoder outputs a feature map. The head is a heatmap decoder with in-channels 1536 and out-channels 308 (keypoints). It uses two deconvolution stages (kernel 4, stride 2) for $4\times$ upsampling, followed by $1\times1$ convolutions with channels $(768, 768, 512)$ and a final $1\times1$ projection to 308 heatmaps. We adopt UDP heatmaps (stride 4, $\sigma{=}6$) and optimize a weighted MSE loss. At test time, we enable flip testing with heatmap fusion.

Optimization uses AdamW (lr $5\times10^{-4}$, $(\beta_1, \beta_2){=}(0.9, 0.999)$, weight decay 0.1) with layer-wise learning-rate decay and zero weight decay for biases, positional embeddings, relative position biases, and norms. We clip gradients to a global $\ell_2$ norm of 1.0. The schedule warms up linearly for 500 iterations (start factor $10^{-3}$), then follows polynomial decay (`power` 1.0) for the remainder. In addition to the main table, we provide fine-grained evaluations in Table 8, which compares Sapiens2 with Sapiens.

| Model | Foot | | Face | | Left Hand | | Right Hand | | Whole Body | |
|---|---|---|---|---|---|---|---|---|---|---|
| | AP | AR | AP | AR | AP | AR | AP | AR | AP | AR |
| Sapiens-0.3B | 72.1 | 77.6 | 82.4 | 86.7 | 66.8 | 72.9 | 67.3 | 73.2 | 70.5 | 77.0 |
| Sapiens-0.6B | 73.8 | 78.9 | 83.9 | 87.8 | 68.4 | 74.1 | 69.0 | 74.5 | 72.8 | 78.6 |
| Sapiens-1B | 75.0 | 80.1 | 85.1 | 88.6 | 69.7 | 75.3 | 70.2 | 75.7 | 74.1 | 79.4 |
| Sapiens-2B | 76.0 | 81.0 | 86.0 | 89.2 | 70.9 | 76.4 | 71.3 | 76.8 | 75.3 | 80.4 |
| Sapiens2-0.4B | 78.4 | 82.0 | 86.2 | 89.5 | 75.1 | 79.0 | 75.6 | 79.4 | 76.9 | 81.3 |
| Sapiens2-0.8B | 80.1 | 83.4 | 87.6 | 90.4 | 76.8 | 80.3 | 77.2 | 80.7 | 79.4 | 83.1 |
| Sapiens2-1B | 81.0 | 84.1 | 88.3 | 90.9 | 77.6 | 81.0 | 78.0 | 81.3 | 80.4 | 84.0 |
| Sapiens2-5B | 82.6 | 85.3 | 89.7 | 91.8 | 79.2 | 82.4 | 79.6 | 82.7 | 82.3 | 85.3 |

Table 8: Pose estimation results on 10K `test` set (K=308). Flip test is used.

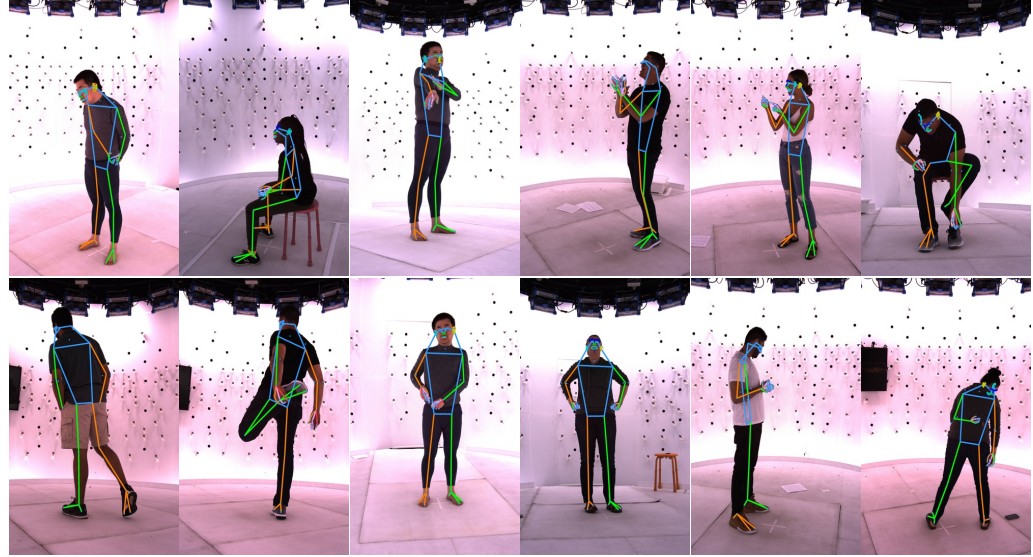

Figure 13: In addition to in-the-wild annotations we also use capture-studio 3D triangulated ground-truth 308 keypoints for finetuning Sapiens2.

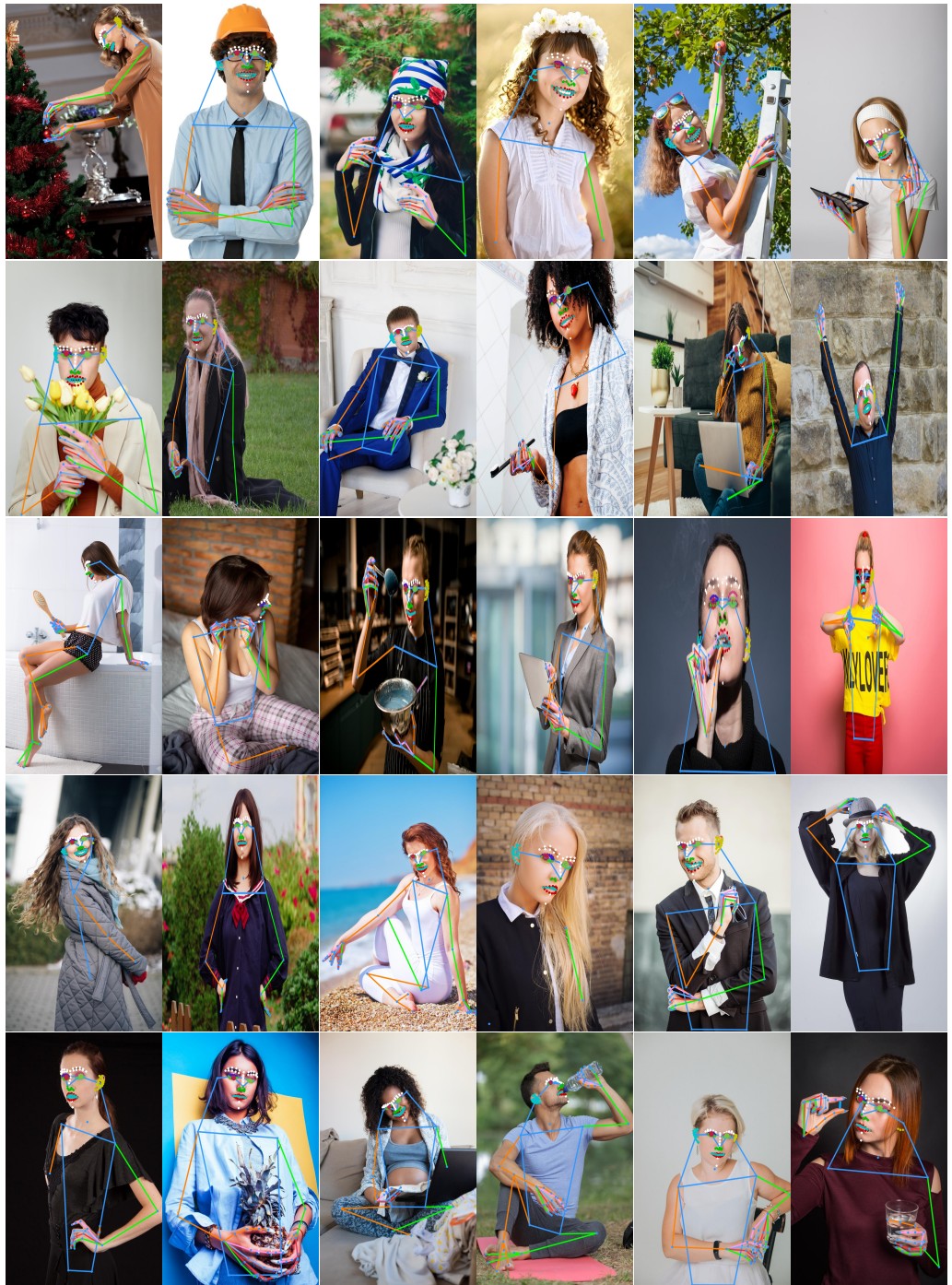

Figure 14: Top-down 308 keypoint predictions using Sapiens2-1B model on in-the-wild images.

## A.3 Body-Part Segmentation

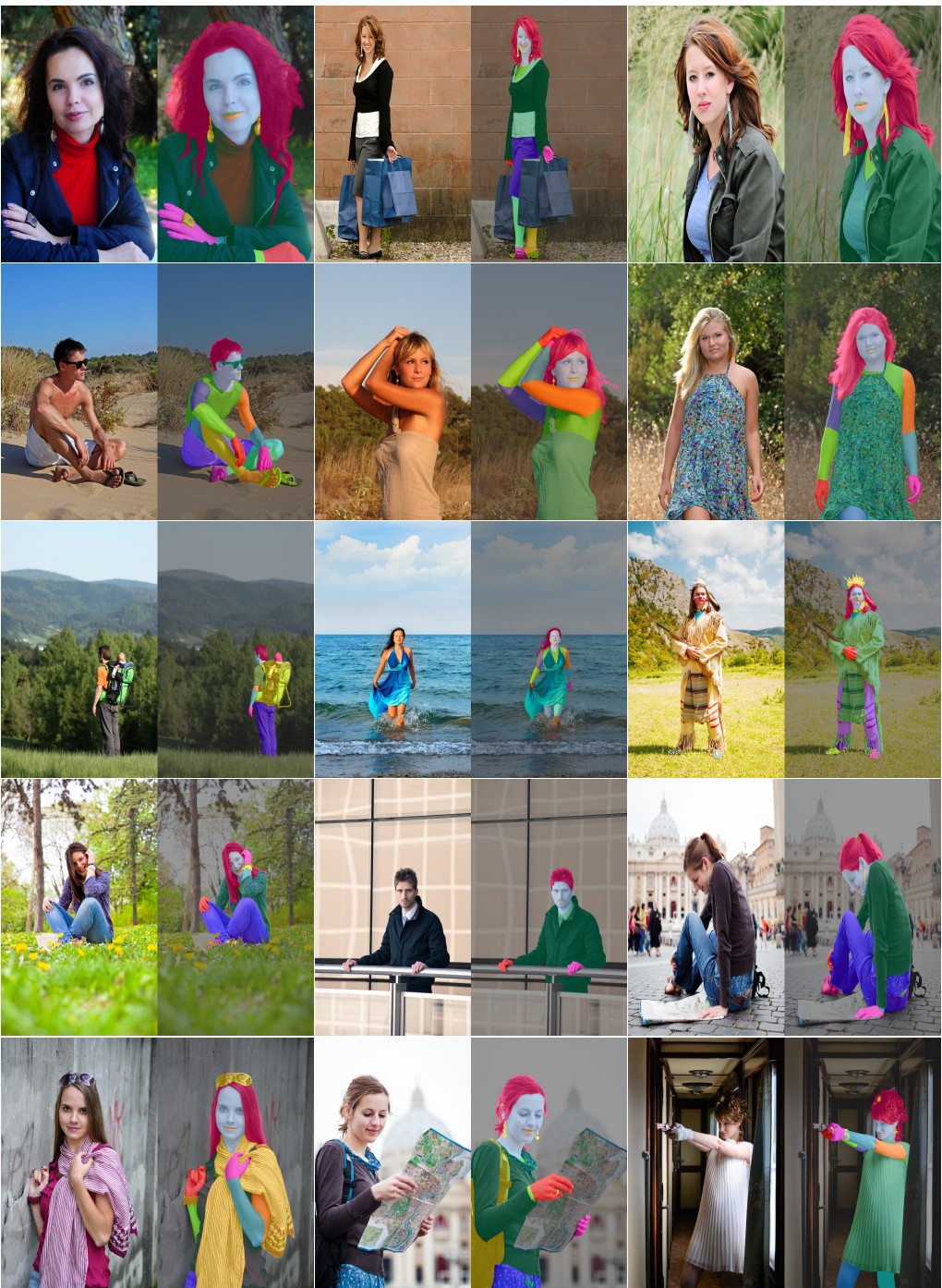

Figure 15: Body-part segmentation (29 classes) using Sapiens2-1B on real-world images.

## A.4    POINTMAP ESTIMATION

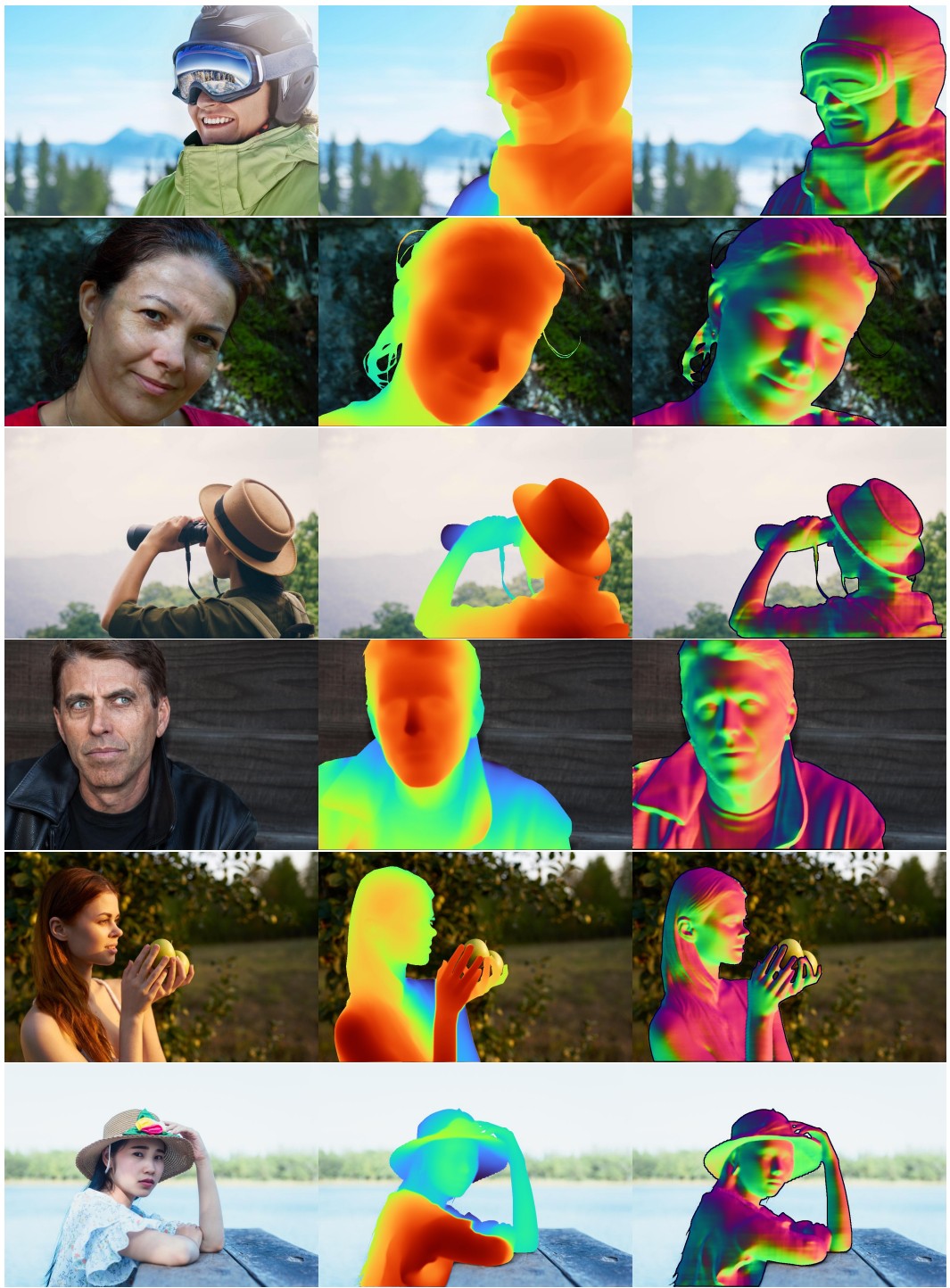

Figure 16: Pointmap using Sapiens2-1B. For each image, we visualize the absolute depth derived from the predicted XYZ pointmap as a heatmap and surface normals computed from depth.

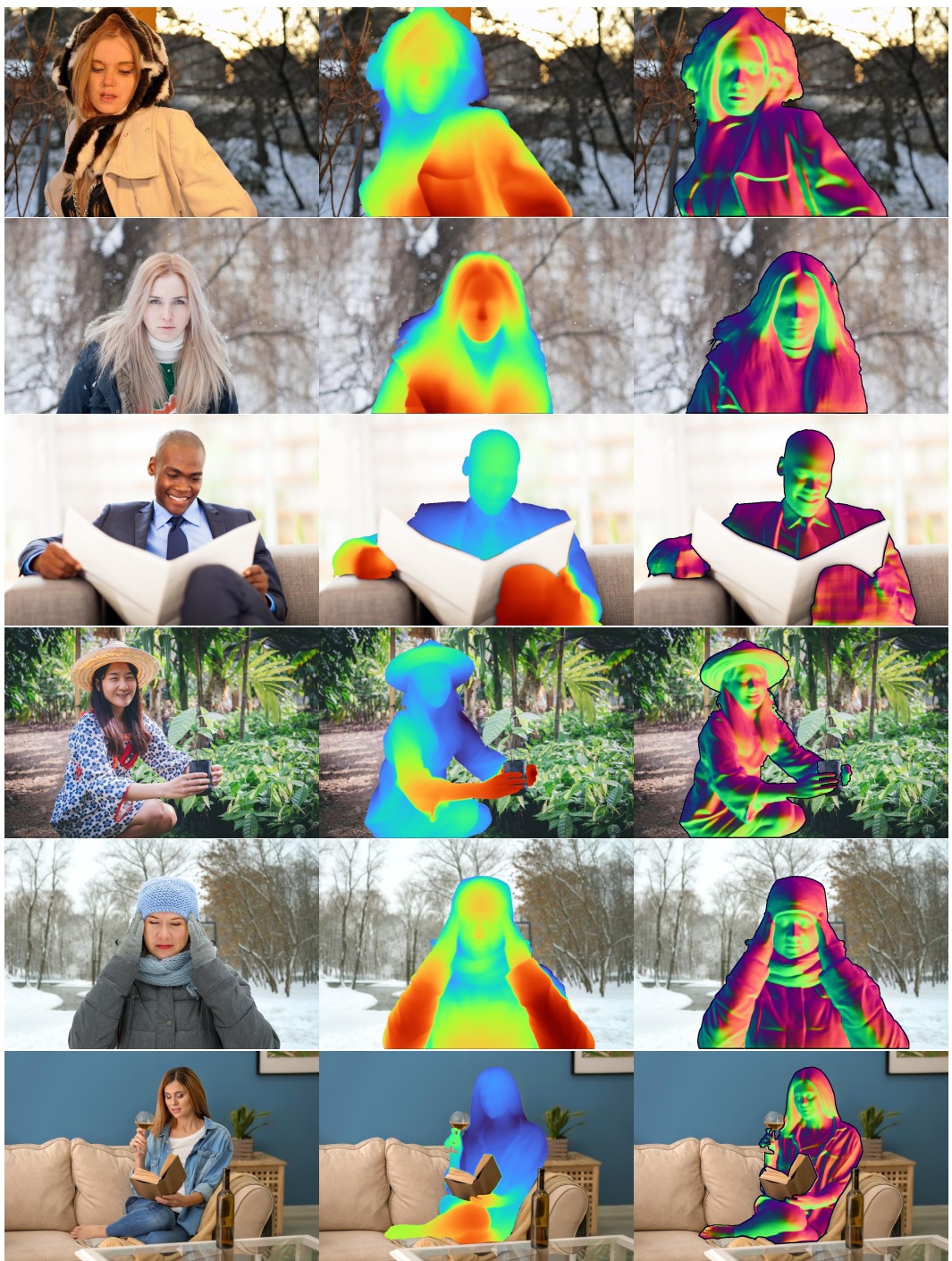

Figure 17: Pointmap using Sapiens2-1B. For each image, we visualize the absolute depth derived from the predicted XYZ pointmap as a heatmap and surface normals computed from depth.

## A.5 NORMAL ESTIMATION

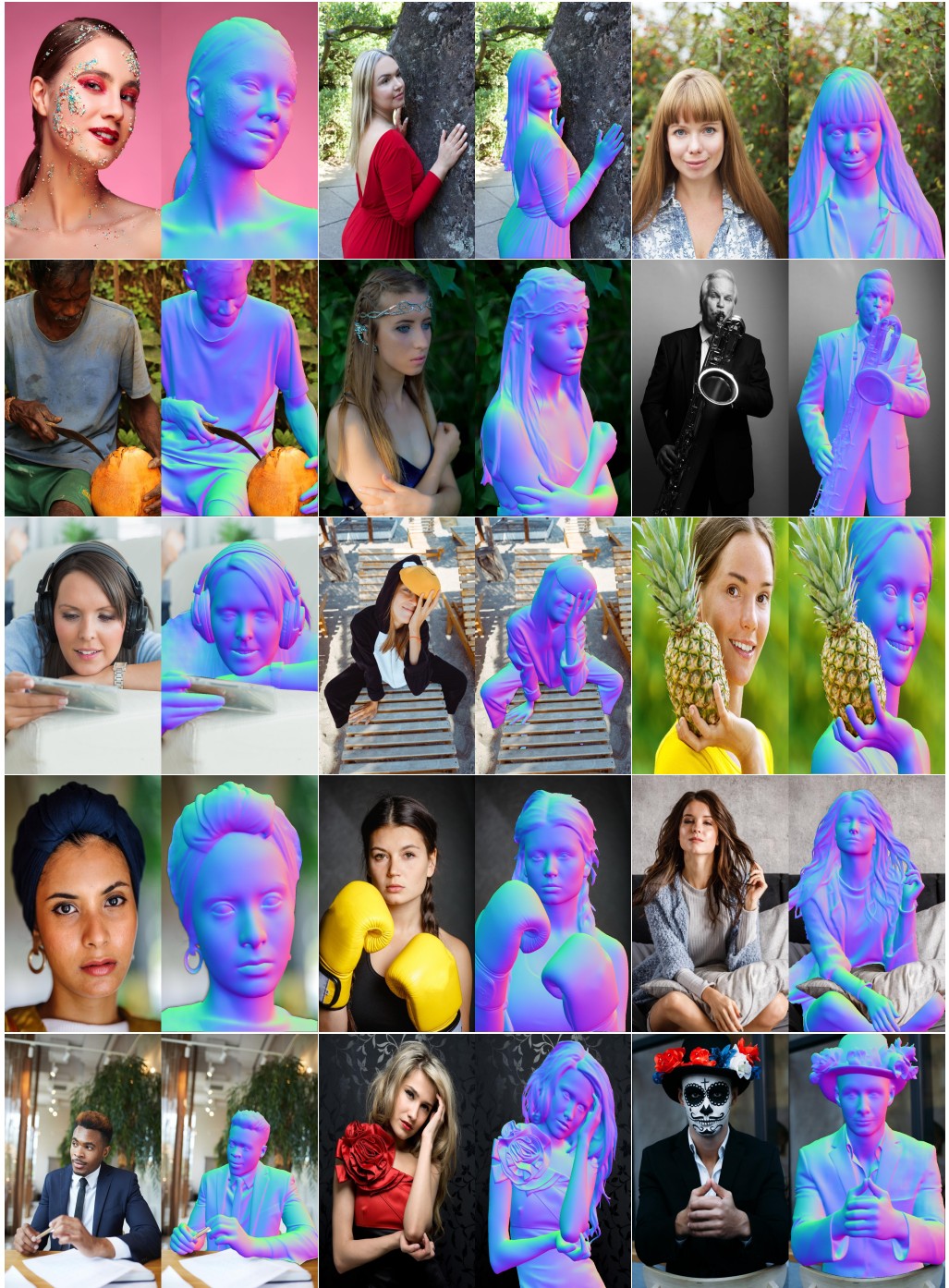

Figure 18: Surface normal prediction using Sapiens2-1B.

## A.6 ALBEDO ESTIMATION

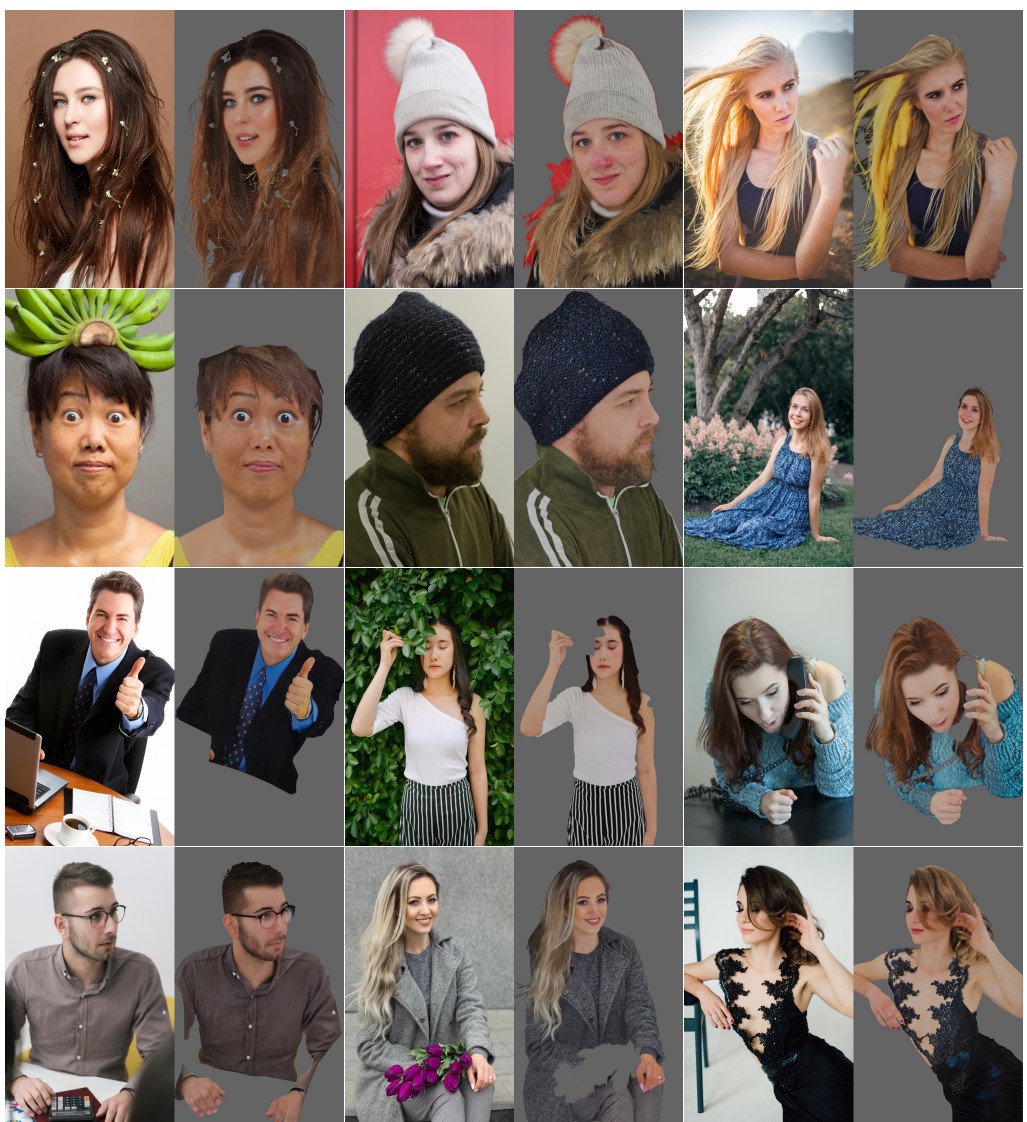

Figure 19: Albedo (base color) prediction using Sapiens2-1B at $1024 \times 768$ resolution.

