# OpenReview forum: "Sapiens2"
_ICLR.cc/2026/Conference — ICLR 2026 Poster_

### Official Review · Reviewer_iUF3 · 2025-10-16

**Soundness:** 3
**Presentation:** 3
**Contribution:** 3
**Rating:** 8
**Confidence:** 4

**Summary:**

The paper presents SAPIENS2, a significant advancement over the original Sapiens foundation model for human-centric computer vision. The work focuses on increasing generalization, versatility, and output fidelity across a broad spectrum of human-related tasks.The core contributions lie in three areas:

1. Introduction of models ranging from 0.4 billion to a 5 billion parameter variant, trained at native 1K resolution, and new hierarchical 4K models capable of 2K output resolution for dense prediction. The Sapiens2-5B model represents one of the highest-FLOPs vision transformers reported.

2. A novel pretraining objective combining Masked Image Reconstruction

3. Scaling the curated pretraining data to 750 million high-quality human images (HUMANS-750M) and increasing post-training task annotations by 10x for tasks including pose, body-part segmentation, depth, normals, and the newly added pointmap and albedo estimation.

**Strengths:**

1. The paper demonstrates compelling gains. The increase in performance over the first generation (e.g., +22.3 mIoU on body-part segmentation and +4 mAP on pose estimation) is substantial and justifies the scale and methodological changes. Crucially, the dense probing results (Table 2) confirm that the pretrained features themselves are vastly superior for human tasks, showcasing strong zero-shot generalization. The Sapiens2-5B model setting a new SOTA with 82.3 mAP on dense 308-keypoint in-the-wild predictions is a major result.

2. The core technical argument—that pure MIM lacks semantic understanding, while CL erodes fine-grained photometric detail—is sound, particularly for high-fidelity human-centric dense tasks (like avatar creation or albedo estimation).

3. The decision to avoid injecting handcrafted human-specific priors (beyond data selection) during pretraining is admirable. This "truly inductive prior-free approach" demonstrates that raw scale and a general-purpose objective can, in this domain, outperform many hand-engineered human-centric models

**Weaknesses:**

No significant weaknesses.

**Questions:**

How does the performance (especially dense vs. semantic tasks) change as $\lambda$ is varied?

Given the "truly inductive prior-free approach" and the massive scale of the Sapiens2-5B model, what is its performance on general computer vision benchmarks that do not involve humans (e.g., COCO object detection, ImageNet linear probing, or ADE20K segmentation)? This would help contextualize Sapiens2's feature learning against general foundation models like DINOv2 and M-ViT at comparable scales.


The paper mentions several architectural stability features (GQA, QK-Norm, RMSNorm, SwiGLU). Which of these, or combination thereof, were most critical for the stable training of the largest 5B parameter, 4K input hierarchical models? Was the training of the largest variant notably more sensitive to hyperparameter choices compared to the smaller models?

---

> ### Author Response · Authors · 2025-11-20
> **Answers to the questions raised in review**
>
> We thank reviewer iUF3 for the positive feedback and suggestions. Please find the answers to the questions raised below:
>
> > How does the performance (especially dense vs. semantic tasks) change as $\lambda$ is varied?
>
> We pretrain Sapiens-0.4b from scratch on 5M randomly selected images using a combined MAE loss and global contrastive (CL) loss with varying weights. Loss = $\lambda$ MAE + $(1-\lambda)$ CL.
> For frozen-encoder evaluation, we use sparse keypoint estimation (COCO-17) as the semantic task and albedo estimation as the dense task.
>
> | lambda | Pose (mAP) | Albedo (MAE, e-2) |
> |:------:|-----------:|--------------:|
> | 0.0 (CL only)   | 36.1       | 6.97         |
> | 0.3    | 38.5       | 6.93          |
> | 0.6 (Ours)   | **40.6**       | 5.44          |
> | 1.0 (MAE only)   | 37.2       | **4.89**          |
>
> For albedo, the MAE objective is most suitable, as it captures low-level detail (best at $\lambda{=}1.0$). For 17-body keypoint estimation, a mixed objective performs best (peak at $\lambda{=}0.6$): CL promotes view-invariance while MAE sharpens localization. Retrieval-based tasks such as re-ID (Fig. 2, main paper) would benefit more from a global CL only objective.
>
> > What is its performance on general computer vision benchmarks that do not involve humans?
>
> Inspired by the feedback, we perform frozen encoder semantic segmentaion on the ADE20k and Cityscapes datasets and report mIoU.
>
> | Model          | ADE20K | Cityscapes |
> |----------------|------:|-----------:|
> | SigLIP2-g       | 42.7  | 64.8       |
> | PE-Spatial-G     | 49.3  | 73.2       |
> | DINOv2-g       | 49.5  | 75.6       |
> | **Ours**    |       |            |
> | Sapiens2-0.8b  | 44.1  | 68.7       |
> | Sapiens2-1b    | 46.3 | 72.0       |
>
> Compared to general backbones of similar capacity, we find that Sapiens features are less suitable for general-vision use cases. This is expected given our model specialization and the large pretraining-to-target distribution shift: ADE20K primarily consists of indoor scenes without human subjects, and Cityscapes comprises autonomous-driving scenes.
>
> > Which of (GQA, QK-Norm, RMSNorm, SwiGLU), or combination thereof, were most critical for the stable training of the largest 5B parameter, 4K input hierarchical models?
>
> We find that adding QK-Norm has the most stabilizing effect when scaling model size and enables longer training. GQA and RMSNorm serve as parameter efficient alternatives to MHSA and LayerNorm, respectively. Lastly, SwiGLU provides higher-capacity feedforward layers compared to standard MLPs, which now is commonly used in many large models (e.g., the LLaMA family).
>
> > Was the training of the largest variant notably more sensitive to hyperparameter choices compared to the smaller models?
>
> For simplicity, we adopt similar hyperparameters across all models, except batch size, learning-rate schedules, and gradient clipping. We set batch size to maximize GPU utilization at each model size. We find that lower learning rates and larger clip values are more suitable for large models during fine-tuning.

---

> > ### Comment · Reviewer_iUF3 · 2025-11-21
> > **Response to authors**
> >
> > Thank you for the rebuttal. You have clarified the key issues, so I decide to keep my score.

---

### Official Review · Reviewer_JoJX · 2025-10-25

**Soundness:** 3
**Presentation:** 3
**Contribution:** 3
**Rating:** 6
**Confidence:** 3

**Summary:**

The paper presents SAPIENS2, a family of high-resolution ViT backbones (0.4B–5B params) for human-centric vision.
The key technical move is a unified pretraining objective that adds a student–teacher contrastive loss on the [CLS] token to MAE reconstruction, aiming to keep low-level fidelity for dense prediction while learning semantic invariances (see Fig. 4 and Sec. 3; the 4K windowed→global hierarchy is sketched in Fig. 5).

The models are trained on a curated ~750M human-image corpus; post-training heads cover pose (308 kp), 29-class part segmentation, pointmap (XYZ), normals, and albedo. Results on the authors’ curated test sets show improvements over the previous SAPIENS-v1 and strong baselines under a frozen-backbone dense-probe protocol (e.g., Tables 2–6; visuals on pages 8–9).

The approach integrates ideas known in the literature—MAE for reconstruction and self-distilled/contrastive training à la DINOv2/iBOT—and is consistent with hierarchical ViTs for high-res inputs (Hiera) and stability tweaks like RMSNorm, QK-Norm, GQA. The paper’s novelty is primarily engineering/integration at scale for human-centric dense tasks rather than introducing a new learning paradigm.

Summary of Review:
The paper introduces a strong unified pretraining framework combining low-level fidelity (MAE) with semantic alignment ([CLS] distillation) in a 4K-ready hierarchical transformer, showing solid technical design and clear scaling trends. However, the lack of public benchmark validation, detailed ablations, and data transparency limits external interpretability. Overall, it’s a well-executed and promising approach but still short of full maturity, justifying a rating of 6 for solid technical depth with room for broader validation.

**Strengths:**

1) Unified pretraining that explicitly marries low-level fidelity (MAE) with semantic alignment ([CLS] distillation); design aligns with robust SSL practice (DINOv2/iBOT/JEPA).
2) 4K-ready hierarchical transformer with practical stability/throughput tweaks (RMSNorm, QK-Norm, GQA) consistent with the literature.
3) Fair frozen-probe protocol across backbones; strong internal scaling trends (0.4B→5B) across pose/seg/geometry (Tables 2–6; pages 8–9).

**Weaknesses:**

1) Public benchmarks missing. Add results on community standards (e.g., COCO-WholeBody 133-kp, plus any public human-part segmentation/normal datasets) to anchor gains.

2) Component ablations. Quantify deltas from RMSNorm vs LayerNorm, QK-Norm, GQA, SwiGLU, and the masking-after-local choice in the 4K pipeline

3) FLOPs claim. Standardize FLOPs reporting (input size, attention type, precision) and compare to published large ViTs (e.g., DINOv2-G, etc) under a common convention.

4) Data governance transparency. Given human images at scale, could be nice to add a data/model card (sources, licenses, demographic coverage, minors handling, opt-out) such that to show overall a general idea regarding the data used.

**Questions:**

1) External benchmarks: Can you report COCO-WholeBody results (with 308→133 mapping) and at least one public normals/segmentation benchmark?
2) More ablations: What are the per-component contributions of RMSNorm / QK-Norm / GQA / SwiGLU and the masking point in 4K?
3) Data disclosures: Confirm the 750M pretraining size, list high-level source categories, and describe license vetting, deduping, and opt-out mechanisms.
4) FLOPs accounting: Please give per-image FLOPs at 1K and 4K, the attention configuration, and compare to DINOv2-G and ViT-22B.
5) Augmentations: How sensitive are albedo/normal metrics to color jitter/solarize on global vs local crops?

Minor:
In Figure 4, I think it should be “L_mae helps the model to learn…” and not “L_mae helps the model learns …”.

---

> ### Author Response · Authors · 2025-11-21
> **Answers to the questions raised in review**
>
> We thank reviewer JoJX for the thoughtful feedback and positive review.
>
> > External benchmarks: Can you report COCO-WholeBody results (with 308→133 mapping) and at least one public normals/segmentation benchmark?
>
> *Pose*: We evaluate on the COCO-WholeBody (COCO-W) val set using the 308→133 mapping on the 114 keypoints common to both formats. For fairness, we re-evaluate ViTPose++ models under the same protocol and report mAP.
>
> | Model | mAP | mAR |
> |:--|--:|--:|
> | ViTPose+-L | 58.9 | 68.1 |
> | ViTPose+-H | **60.1** | **69.4** |
> | Sapiens2-0.4B | 54.3 | 62.8 |
> | Sapiens2-0.8B | 56.0 | 64.5 |
> | Sapiens2-1B | 57.9 | 66.9 |
>
> While specialized models like ViTPose++ trained explicitly on COCO-W achieve slightly higher metrics, Sapiens2 achieves competitive performance zero-shot, without seeing COCO-W training data. This holds despite imperfect 1:1 keypoint-definition alignment (e.g., hip, fingertip, and shoulder) and while predicting a substantially larger set of 308 keypoints.
>
> *Normal*: Following Sapiens, ECCV 2024, we evaluate normal prediction on the Hi4D dataset and compare with DaViD, ICCV 2025.
> Our models outperform prior work and set a new state of the art on Hi4D for normal estimation.
>
> | Model | Angular Error Mean | % within 11.5 deg |
> |:--|--:|--:|
> | DaViD-Base | 15.72 | 43.22 |
> | DaViD-Large | 15.37 | 45.11 |
> | Sapiens-1B | 12.18 | 60.36 |
> | Sapiens-2B | 12.14 | 60.22 |
> | Sapiens2-1B | 10.69 | 67.79 |
> | Sapiens2-5B | **9.43** | **69.12** |
>
>
> > More ablations: What are the per-component contributions of RMSNorm / QK-Norm / GQA / SwiGLU and the masking point in 4K?
>
> Thank you for the suggestion. Since full ablations at our pretraining scale are computationally expensive, we defer to the respective publications for per-component contributions - QK-Norm (Chameleon, 2024) and RMSNorm, GQA (LLaMA, 2023). Inspired by the feedback, we ablate the addition of SwiGLU (vs. a standard MLP) by pretraining Sapiens2-0.4B model from scratch on 5M randomly selected images and performing dense probing evaluations on pose and segmentation, keeping the embedding dimension fixed.
>
> | Backbone | Pose (mAP) | Seg (mIoU) |
> |:------:|-----------:|--------------:|
> | Base   |  14.6     |     21.7     |
> | SwiGLU    |  18.9  |   24.5        |
>
> The table shows that SwiGLU increases the capacity of the feedforward blocks, improving feature learning and yielding gains of 4.3 mAP and 2.8 mIoU on our shallowest backbone.
>
> We evaluate the masking strategy at 4K (4096x3072) resolution by comparing independent 16×16 patch drops to dropping 2×2 patch blocks (ours), which induces a global 64×64 drop. Our pretraining and dense-probing setups are the same as above, except for the increased resolution. To maintain a 0.75 masking ratio, under independent masking we drop the pooled global token after the local stage whenever more than half of the local tokens are masked.
>
> | Masking | Pose (mAP) | Seg (mIoU) |
> |:------:|-----------:|--------------:|
> | Independent   |   12.5   |    18.4      |
> | Block (Ours)    |  21.1  |   28.9        |
>
> Since a 16x16 patch at 4K covers only a tiny fraction of image (0.4% pixels), masking at this granularity in our hierarchical setup harms global patch-level context.
>
> > Data disclosures: Confirm the 750M pretraining size, list high-level source categories, and describe license vetting, deduping, and opt-out mechanisms.
>
> Yes, the 750M images are sourced from (i) licensed third-party vendors with permissible rights and (ii) a proprietary corpus with appropriate opt-out mechanisms. All of the data is extensively filtered and de-duplicated using practices similar to DINOv2.
>
> > FLOPs accounting: Please give per-image FLOPs at 1K and 4K, the attention configuration, and compare to DINOv2-G and ViT-22B.
>
> As requested, we compare the FLOPs of all the backbones at their recommended image resolutions.
> | Method         | #Params | TFLOPs | Image size |
> |:--------------|--------:|------:|-----------:|
> | DINOv2-G      | 1 B     | 0.29  | 224        |
> | ViT-6.5B      | 6.5 B   | 1.66  | 224        |
> | ViT-22B       | 22 B    | 11.36 | 224        |
> | Sapiens2-0.4B | 0.4 B   | 1.26  | 1024       |
> | Sapiens2-0.8B | 0.8 B   | 2.59  | 1024       |
> | Sapiens2-1B   | 1 B     | 4.72  | 1024       |
> | Sapiens2-1B-4K| 1 B     | 6.43  | 4096       |
> | Sapiens2-5B   | 5 B     | 15.72 | 1024       |
>
> > Augmentations: How sensitive are albedo/normal metrics to color jitter/solarize on global vs local crops?
>
> In general, we find that downstream metrics that depend on low-level color cues—for example, albedo estimation—are negatively impacted by the color invariance induced by photometric augmentations such as color jitter and solarize, leading to a degradation of roughly 5 dB in PSNR.

---

> ### Comment · Reviewer_JoJX · 2025-11-23
> **Response to authors**
>
> Thank you for the clean response and rebuttal. You have addressed my concerns and issues so I am inclined to increase my score.

---

### Official Review · Reviewer_mhqq · 2025-10-28

**Soundness:** 3
**Presentation:** 3
**Contribution:** 3
**Rating:** 8
**Confidence:** 5

**Summary:**

This paper introduces Sapiens2, a family of high-resolution vision Transformer models ranging from 0.4B to 5B parameters designed for human-centric vision tasks. It aims to achieve stronger generalization, task versatility, and high-fidelity output. Building upon its predecessor Sapiens, Sapiens2 undergoes comprehensive upgrades across three dimensions: pre-training, data, and architecture.

Unified Pre-training Objective: It combines Masked Image Modeling (MAE) with Self-Distilled Contrastive Learning (CL). This approach preserves pixel-level details beneficial for dense prediction tasks while learning high-level semantics beneficial for zero-shot/few-shot generalization. The joint objective avoids the limitations of pure MAE which lacks semantics and pure CL which neglects details.

Large-Scale, High-Quality Human-Centric Data: The Humans-750M dataset is constructed containing 750 million high-quality human images filtered through multiple stages. It covers diverse populations, poses, scenes, and lighting conditions. The only requirement is that each image contains at least one prominent person, with no other manual priors.

High-Resolution Scalable Architecture: The model supports native 1K resolution and pioneers a 4K resolution hierarchical model achieved through window attention plus global attention. It integrates cutting-edge techniques including RMSNorm, Grouped-Query Attention (GQA), QK-Norm, SwiGLU FFN, and a PixelShuffle decoder to enhance training stability and efficiency. The pre-training output resolution reaches 2K, significantly improving detail representation for dense predictions.

Extensive Downstream Task Coverage and SOTA Performance: After pre-training, the model achieves state-of-the-art results across five dense prediction tasks including pose estimation, body part segmentation, depth estimation, surface normal estimation, and albedo estimation. It substantially outperforms its predecessor and existing methods, demonstrating significant improvements such as pose estimation mAP improved by +4, body segmentation mIoU improved by +22.3, and normal estimation angular error reduced by 29.2%.

**Strengths:**

1. This paper adopts a simple joint pre-training paradigm of "masked reconstruction + contrastive learning." Although this technique has been widely used in various fields, its application in the human-centric vision domain effectively balances low-level detail and high-level semantic learning, addressing the trade-off between dense prediction and semantic understanding in existing self-supervised methods. Additionally, it scales the Vision Transformer to 4K resolution while supporting 2K output.

2. The experiments are exceptionally solid. Not only does it establish new SOTA results across multiple tasks, but it also demonstrates the model's strong generalization capabilities in zero-shot, few-shot, and fully supervised scenarios through dense probing, ablation studies, and extensive visualizations (e.g., Figures 1, 7–10). The descriptions of data, architecture, and training strategies are detailed and reproducible.

3. A large-scale human-centric dataset was collected and used for training, validating the effectiveness of large-scale data and large models in the human vision domain.

4. The provision of a series of models of different sizes can significantly advance industrial applications.

**Weaknesses:**

1. Limited generalization in crowded scenes: As mentioned on page 9 of the paper, the model performs excellently on images containing 1-4 prominent people but shows performance degradation in crowded scenes or complex multi-agent interactions. This limits its application in real-world scenarios such as surveillance and sports events. Future work needs to explore more robust multi-person modeling mechanisms.

2. Synthetic data discussion: For geometric and material tasks like point maps, normals, and albedo, training relies entirely on synthetic data. Although results show good generalization (Figure 10), the synthetic-real domain gap remains a potential risk.

3. Data processing pipeline details: The data processing pipeline is important in this paper. While the paper mentions the data processing pipeline, details of each step - such as specific annotation verification and sampling strategies - could be provided. Alternatively, forming a methodology for the community would promote development.

**Questions:**

1. After collecting such a large volume of human-centric data, what instructive conclusions can be drawn? Building upon a decent pre-trained model, can we continue to scale up the data volume, and what type of additional data would be more effective?

2. When real data is exhausted, considering the use of synthetic data—such as leveraging AIGC models to generate human-centric data—would such generated data be beneficial for this type of discriminative model?

---

> ### Author Response · Authors · 2025-11-20
> **Answers to the questions raised in review**
>
> We thank reviewer mhqq for their positive remarks and thoughtful review. We answer the question below:
>
> > After collecting such a large volume of human-centric data, what instructive conclusions can be drawn?
>
> We have three insights that we expect to benefit projects at similar scale; these also align with established findings in language modeling.
>
> i) Balanced, higher-quality, smaller amounts of data are more helpful than larger-scale raw data. This aligns with existing works such as DeepSeek-v3 and DINO-v2. Simple processing can significantly amplify the benefits of scale.
>
> ii) Matching the pretraining data distribution to the test-time distribution is very helpful. If you know the target distribution in advance—for example, humans in low-light settings or egocentric camera views—enrich or replace the pretraining data accordingly to benefit from unsupervised representation learning. However, this can reduce generalization to other use cases.
>
> iii) Curriculum over pretraining data: when ordering data or using multi-stage training, place the highest-quality data at the end.
>
> > Building upon a decent pre-trained model, can we continue to scale up the data volume?
>
> Yes. The vision models and data are an order of magnitude smaller than frontier language models. We continue to see benefits from scaling up data volume. However, it is crucial to scale model size appropriately as well.
>
> > What type of additional data would be more effective?
>
> Sapiens family is trained primarily on images with prominent human presence, typically featuring 1–4 people. To improve generality across the human domain, especially for multi-human scenarios and crowded scenes, additional human-centric data that is more diverse than the current pretraining corpus would yield step-change gains.
>
> > When real data is exhausted, considering the use of synthetic data—such as leveraging AIGC models to generate human-centric data—would such generated data be beneficial for this type of discriminative model?
>
> Assuming the suggestion is to use AIGC-generated data in Sapiens pretraining (please correct us otherwise), this is a promising direction. Synthetic data can improve domain coverage, and we plan to explore it in future work.

---

> > ### Comment · Reviewer_mhqq · 2025-11-24
> > **response**
> >
> > Thank you to the author for the solid experiments and valuable insights provided. I will maintain my positive rating.

---

### Official Review · Reviewer_my7v · 2025-10-28

**Soundness:** 3
**Presentation:** 3
**Contribution:** 3
**Rating:** 6
**Confidence:** 3

**Summary:**

The paper introduces Sapiens2, a new family of Vision Transformer models ranging in size from 0.4 billion to 5 billion parameters. This model family constitutes the second generation of the Sapiens line (ECCV 2024), maintaining the original goal of creating versatile foundation models centered on inputs/outputs depicting humans. The core focus is on achieving strong generalization, versatility, and high-fidelity outputs.

The primary differences from the original Sapiens models lie in the larger scale of the training data, the pretraining paradigm (hybrid instead of MAE), and the supported resolution (4K). Specifically, Sapiens2 models are pretrained on a dataset of 750 million human-centric images, isolated and curated from an initial web-scale pool of approximately 4 billion images. The pretraining employs a hybrid self-supervised paradigm that combines Masked Auto-Encoding (MAE) with semantic contrastive learning, a technique previously explored by methods like CMAE (Huang et al., TPAMI 2022) for image classification.

The models are evaluated across five human-centric tasks: pose estimation, body-part segmentation, depth estimation, surface normals estimation, and albedo estimation. Fine-tuning for each task involves adding and training a lightweight task-specific head while keeping the large backbone frozen.

**Strengths:**

1. State-of-the-Art Performance: The models demonstrate state-of-the-art results across all five evaluated tasks, showcasing the efficacy of the scaling and pretraining approach.
2.  The evaluation is comprehensive, covering five complex, pixel-level tasks with strong quantitative results and compelling qualitative examples, effectively demonstrating the models' high-fidelity capabilities.
3. The paper is well written and well structured
4. Contribution of high resolution (up to 4K) specialized foundation models focused on complex human analysis (conditional on release).

**Weaknesses:**

1. Limited Fundamental Novelty: the primary advances appear to be an increase in scale (data and backbone size) and the application of an existing hybrid self-supervised paradigm (MAE + contrastive learning, previously seen in classification) to a new domain (human-centric generation). The paper does not introduce novel fundamental insights or methodological breakthroughs in transformer architecture or pretraining theory.
2. It is currently unclear whether the dataset, code, or the trained model checkpoints will be publicly released. Given that the main contribution of the paper resides in the achieved performance and the massive scale of the trained model family, the lack of public availability would severely limit the reproducibility, impact, and overall contribution to the research community.

**Questions:**

1. Are the authors planning to publicly release the curated 750M image dataset, the source code for the pretraining and fine-tuning pipelines, and/or the trained Sapiens2 model weights?

2. Ablation on Pretraining: Could the authors provide an ablation comparing the hybrid self-supervised objective (MAE + Contrastive) against the individual components (MAE-only and Contrastive-only) to justify the added complexity and quantify the specific performance gain attributed to the combined loss?

---

> ### Author Response · Authors · 2025-11-22
> **Answers to the questions raised in review**
>
> We thank reviewer my7v for the feedback and suggestions. Please find the answers to the questions raised below:
>
> > Are the authors planning to publicly release the curated 750M image dataset, the source code for the pretraining and fine-tuning pipelines, and/or the trained Sapiens2 model weights?
>
> Yes, we are committed to open-sourcing and will release all our pretrained and fine-tuned models, along with the code, to facilitate future research. On data, due to privacy concerns, we will not be able to open-source the images.
>
> > Ablation on Pretraining: comparing the hybrid self-supervised objective (MAE + Contrastive) against the individual components (MAE-only and Contrastive-only)
>
> We study the contributions of global contrastive loss and masked autoencoding by pretraining Sapiens2-0.4B from scratch on 5M randomly selected images and performing dense probing on Pose-308, segmentation-29, and albedo estimation.
>
> | Loss | Pose (mAP) | Seg (mIoU) | Albedo (mAE, e-2) |
> |:------:|-----:|--------------:|--------------:|
> | CL   |  13.3   |  22.5 | 6.97         |
> | MAE   |  16.8 |  23.1  | **4.89**          |
> | MAE + CL   |  **18.9** |  	**24.5**   | 5.44          |
>
> MAE-only models perform best on tasks that rely on low-level, color-based cues (e.g., albedo recovery), indicating that the learned representations encode fine image details. By contrast, the contrastive loss introduces view invariance and, when combined with MAE, improves semantic reasoning—benefitting pose estimation and segmentation (e.g., distinguishing left from right attributes). We will modify the manuscript to include this ablation.

---

> > ### Comment · Reviewer_my7v · 2025-11-28
> >
> > Thank you for addressing my questions. I am now inclined to raise my score given the rebuttal and the thorough response to the concerns raised by other reviewers.

---

### Meta-Review · Area_Chair_aKWn · 2025-12-16

**Summary:**

All the reviewers give the positive feedback with the average rating of 7.00. Most of concerns are about the experimental analysis, which the authors have addressed well. I agree with the reviewers and decide to accept the paper as a poster. Considering the open-sourcing of the model and code in the form of visual foundation model, an acceptance of oral is also appropriate.

**Reviewer Concerns:**

I think most concerns are well addressed.
The experiments the reviewers cared about most, for example, the individual roles of CL and MAE, are solid and convincing, which makes two reviewers of relatively lower score decide to increase the score.

**Reviewer Scores:**

The average rating is initially 7.0, a score worthy of acceptance. Meanwhile, the Reviewer my7v and Reviewer JoJX promised to increasing the score while the other two maintained the score. After reading the discussion, I am inclined to believe the motivations of the reviewers are reasonable.

---

### Decision · Program_Chairs · 2026-01-26

Accept (Poster)